# Detached Skip-Links and $R$-Probe: Decoupling Feature Aggregation from Gradient Propagation for MLLM OCR

Ziye Yuan [*1]   Ruchang Yao [*2]   Chengxin Zheng [3]   Yusheng Zhao [1]   Daxiang Dong [†3]   Ming Zhang [†1]

## Abstract

Multimodal large language models (MLLMs) excel at high-level reasoning yet fail on OCR tasks where fine-grained visual details are compromised or misaligned. We identify an overlooked optimization issue in multi-layer feature fusion. Skip pathways introduce direct back-propagation paths from high-level semantic objectives to early visual layers. This mechanism overwrites low-level signals and destabilizes training. To mitigate this gradient interference, we propose Detached Skip-Links, a minimal modification that reuses shallow features in the forward pass while stopping gradients through the skip branch during joint training. This asymmetric design reduces gradient interference, improving stability and convergence without adding learnable parameters. To diagnose whether fine-grained information is preserved and usable by an LLM, we introduce $R$-Probe, which measures pixel-level reconstructability of projected visual tokens using a shallow decoder initialized from the first quarter of the LLM layers. Across multiple ViT backbones and multimodal benchmarks, and at scales up to 7M training samples, our approach consistently improves OCR-centric benchmarks and delivers clear gains on general multimodal tasks.

## 1. Introduction

Multimodal large language models (MLLMs) have rapidly advanced the integration of vision and language, showing strong performance in high-level semantic reasoning and dialogue.(Team et al., 2023; Achiam et al., 2023; Wang et al., 2025; Bai et al., 2025) However, they still exhibit a clear performance gap on low-level perception tasks, particularly Optical Character Recognition (OCR) and fine-grained visual grounding. Existing benchmarks show that even state-of-the-art models often hallucinate text in dense documents or fail to resolve small objects in high-resolution scenes(Fu et al., 2024; Liu et al., 2024; Kanade & Ganu, 2025).

Previous studies identify the pre-trained Vision Transformer (ViT) as a primary bottleneck (Liu et al., 2025a). By design, contrastive objectives (e.g., CLIP) encourage semantic alignment, pushing the encoder to abstract away spatial details to align with global text descriptions (Tong et al., 2024; Bolya et al., 2025). To mitigate this information loss, recent approaches either incorporate auxiliary objectives such as reconstruction losses (Fini et al., 2025; Tschannen et al., 2025), or rely on end-to-end generative supervision from the multimodal LLM through its next-token prediction (NTP) loss (Guo et al., 2024; Chen et al., 2024b). In parallel, architectural designs increasingly adopt multi-layer fusion to incorporate shallow features that retain geometric and pixel-level information (Yao et al., 2024; Wei et al., 2024; Lin et al., 2025). While intuitively promising, we identify a previously under-discussed optimization issue in this fusion-based architectures. Straightforward fusion establishes direct backpropagation paths from the LLM's semantic objectives to early visual blocks. This subjects shallow layers originally optimized for low-level patterns to conflicting high-level supervision, resulting in gradient interference and training instability, as in Figure 2.

To address this trade-off between spatial detail and training stability, we propose Detached Skip-Links (Fig. 1). By stopping gradients on shallow features before fusion, we decouple feature aggregation from gradient optimization. This mechanism passes fine-grained visual details to the LLM while preventing semantic-heavy gradients from destabilizing shallow layers, without sacrificing the simplicity of skip-connections. Both theoretical analysis and empirical results show that this operation reduces gradient conflict, leading to improved training stability and faster convergence. These effects are consistent across different ViT backbones

---

[*]Equal contribution (random order).   [†]Corresponding authors. [1]State Key Laboratory for Multimedia Information Processing, School of Computer Science, PKU-Anker LLM Lab, Beijing Key Laboratory of Software and Hardware Cooperative Artificial Intelligence Systems, Peking University, Beijing, China [2]Tsinghua University, Beijing, China [3]Baidu Inc, Beijing, China. Correspondence to: Ming Zhang <fmzhang_cs@pku.edu.cn>, Daxiang Dong <dongdaxiang@baidu.com>.

*Proceedings of the $43^{rd}$ International Conference on Machine Learning*, Seoul, South Korea. PMLR 306, 2026. Copyright 2026 by the author(s).

and remain stable when scaling training to 7M samples.

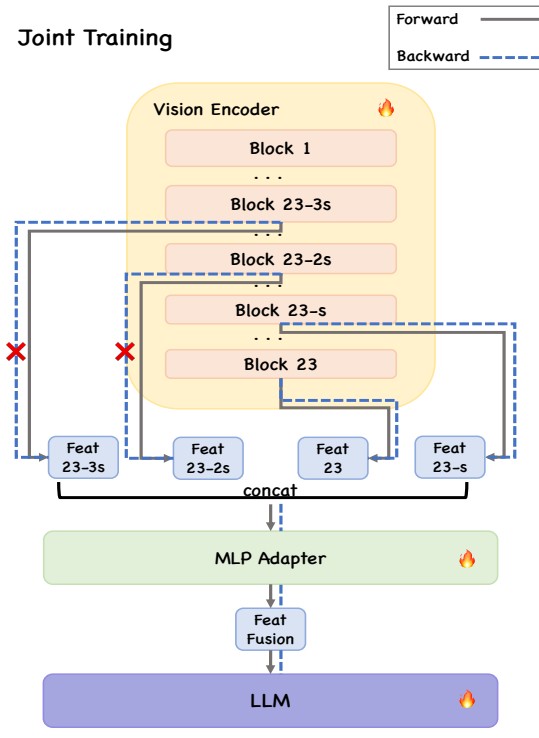

*Figure 1.* **Overview of Detached Skip Links.** Intermediate features are concatenated with the final output along the channel dimension, here $S$ denotes stride. We apply a stop-gradient operation to shallow skip features before fusion. This design effectively delivers fine-grained details to the LLM while shielding early layers from optimization conflicts.

Beyond optimization, a major challenge in improving fine-grained perception is the lack of a reliable diagnostic metric. Standard downstream benchmarks are often noisy proxies for visual capability, as MLLMs can bypass perception by exploiting language priors or parametric knowledge(He et al., 2025). Existing probing methods typically use linear classifiers(Pagh et al., 2007; Alain & Bengio, 2016; Dosovitskiy, 2020) to assess representation quality, but such tasks are insufficient for measuring fine-grained visual information. To systematically quantify the visual signal effectively transmitted to the LLM, we introduce the Reconstruction Probe ($R$-Probe). Moving beyond traditional linear separability tests, $R$-Probe assesses the recoverability of visual details through pixel-level reconstruction. By initializing the reconstruction head with the first quarter layers of the target LLM, we simulate the actual information injection process, probing features exactly as they are "perceived" by the language model. We posit that minimal reconstruction loss from the projected tokens serves as a proxy for effective information preservation.

Our main contributions are summarized as follows:

- **Introduce Detached Skip-Links**: reuse shallow ViT features for fusion while stopping gradients through the selected skip branch, reducing gradient interference and improving training stability without additional parameters.

- **Propose $R$-Probe**: a reconstruction-based diagnostic that measures whether projected visual tokens retain fine-grained information and remain directly decodable by an LLM-initialized shallow decoder.

- **Demonstrate effectiveness at scale**: extensive experiments across ViT backbones and 22 benchmarks show strong gains on OCR-centric tasks and consistent improvements on general multimodal evaluation.

**Conflict of Interest Disclosure.** The authors declare that they have no financial conflicts of intereset related to this work.

## 2. Related Work

**OCR-Centric Multimodal Large Language Models.** Recent advances have shifted from modular OCR pipelines to end-to-end MLLMs that internalize text recognition. While some systems (Cui et al., 2025b) focused on localized extraction, recent large-scale MLLMs (Chen et al., 2024a; Lu et al., 2024; Wu et al., 2024b; Cui et al., 2025a; Team et al., 2025a;b; Bai et al., 2025; Dong et al., 2025; Wang et al., 2025) demonstrate that massive pretraining with OCR-synthesized data yields strong character-level capabilities. To capture fine-grained details, modern architectures often employ dynamic resolution or image-splitting strategies (Chen et al., 2024a; Bai et al., 2025). However, effectively injecting these high-fidelity features into the LLM context remains an open challenge. While earlier general-purpose models relied on heavy bridging modules like (Alayrac et al., 2022; Li et al., 2023; Guo et al., 2024), recent OCR-specific approaches favor multi-scale fusion (Ye et al., 2023; Li et al., 2024) or deep cross-attention (Bai et al., 2025; Chen et al., 2026). Representative examples include TextHawk (Yu et al., 2024), with token compression and multi-level cross-attention, and VLM-FO1 (Liu et al., 2025b), which strengthens regional OCR using auxiliary high-resolution encoders and explicit region tokens. In contrast to complicated architectures, our work provides a lightweight training-time solution without introducing new modules, and is orthogonal to these architectural designs.

**Multi-layer Fusion Strategies.** To mitigate information loss in deep ViTs, researchers have explored leveraging intermediate features. Methods like DeepStack (Meng et al., 2024), DenseConnector (Yao et al., 2024) and (Gao et al., 2022; Lin et al., 2025) aggregate representations from multiple depths, while others combine signals from distinct vi-

sion backbones (Shi et al., 2024; Kar et al., 2024; Wu et al., 2024b) or use hierarchical schemes (Zhang et al., 2024). While conceptually sound, such heterogeneous fusion often exhibits unstable optimization dynamics in practice. In the broader optimization literature, controlling or blocking gradient flow has been explored as a strategy to improve training stability in settings such as recurrent networks and representation learning (Arpit et al., 2018; Yu et al., 2020). A plausible source of instability is the interference between high-level semantic gradients from the LLM and the shallow layers' role in preserving fine-grained visual details, such as character strokes. As a result, naive fusion can lead to gradient conflicts and impair training stability. We therefore investigate whether decoupling feature propagation from gradient flow can alleviate such interference, while retaining the benefits of shallow visual cues.

**Detail Preservation and Diagnostics.** Several works preserve or recover fine detail by introducing specialized tokens and reconstruction/decoding objectives, including AURORA/Perception Tokens (Bigverdi et al., 2025), Morph-Tokens (Pan et al., 2024) and SeTok (Wu et al., 2024a). Our $R$-Probe is aligned in spirit as a fidelity assessment, but differs by being probe-only. Specifically, it reuses a shallow decoder initialized from the quater of LLM layers to probe representations, without introducing additional heavyweight decoding modules.

# 3. Detached Skip Links

## 3.1. Motivation

Skip-based multi-scale fusion is a common strategy for injecting fine-grained visual details into vision–language models. However, straightforward skip connections introduce a direct gradient path from the language modeling objective to shallow visual blocks. These shallow blocks primarily encode low-level geometric structures, which differ from the high-level semantic objectives of the LLM.

Empirically, we observe that allowing full gradient backpropagation through skip connections leads to diffused and structurally inconsistent attention patterns in shallow layers (Fig. 2). The visualization follows prior work (Darcet et al., 2024), with implementation details provided in the appendix A.

As shown in Fig. 2 (*Middle*), gradients dominated by semantic objectives disrupt pre-trained spatial priors, degrading the encoder's ability to localize fine-grained features. In contrast, detaching gradients (Fig. 2, *Right*) preserves baseline-like structural consistency. This observation motivates **Detached Skip-Links**, a mechanism designed to decouple feature aggregation from gradient propagation.

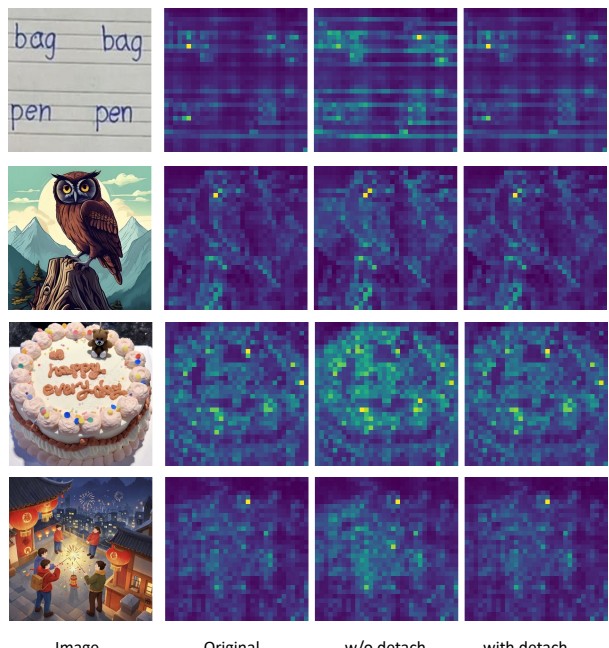

Image     Original     w/o detach     with detach

*Figure 2.* **Impact of gradient backpropagation on shallow-layer representations.** We visualize [CLS] attention maps of the 4th ViT block. *Left*: Original frozen attention patterns. *Middle* (**Full Gradients**): Backpropagation from the LLM leads to diffused and structurally inconsistent attention, as semantic-heavy gradients disrupt pre-trained spatial priors. *Right* (**Detached**): Detaching gradients preserves fine-grained structural consistency.

## 3.2. Architecture

To resolve the conflict described above, we adopt a selective detachment strategy based on feature depth. To formalize this, we partition the intermediate skip features into two sets: a shallow group $\mathbf{h}_{\text{shallow}}$ (e.g., blocks 6, 12) and a deep group $\mathbf{h}_{\text{deep}}$ (e.g., blocks 18, 23). The input to the fusion adapter is constructed as:

$$\mathbf{z} = \text{MLP}\left(\left[\mathbf{h}_{\text{main}}; \mathbf{h}_{\text{deep}}; \text{sg}(\mathbf{h}_{\text{shallow}})\right]\right), \qquad (1)$$

where $[\cdot]$ denotes concatenation and $\text{sg}(\cdot)$ is the stop-gradient operator.

This selective detachment establishes a dual-pathway optimization landscape, grounded in the hypothesis that feature depth dictates optimization compatibility. Deep features inherently align with the output and benefit from joint optimization. In contrast, shallow features primarily capture low-level geometry and are more susceptible to distortion under direct supervision. By allowing gradients to backpropagate only through the deep branch, our design enables semantic alignment while simultaneously shielding early layers from interference. This ensures that low-level cues remain robust and fully available for the forward pass (as supported by Proposition 4.1). Comprehensive ablations an-

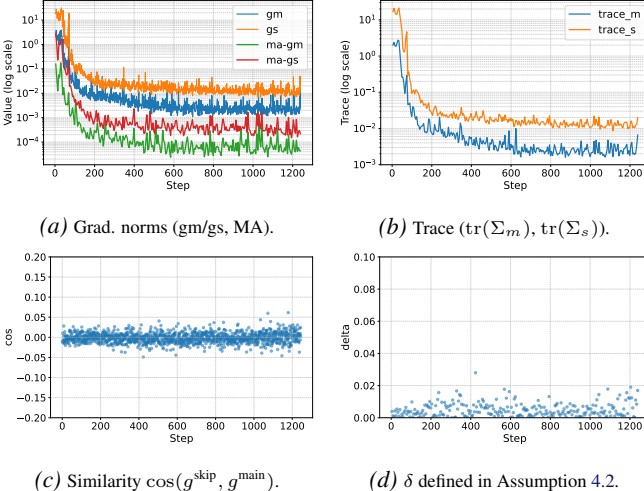

*(a)* Grad. norms (gm/gs, MA).

*(b)* Trace ($\mathrm{tr}(\Sigma_m)$, $\mathrm{tr}(\Sigma_s)$).

*(c)* Similarity $\cos(g^{\mathrm{skip}}, g^{\mathrm{main}})$.

*(d)* $\delta$ defined in Assumption 4.2.

*Figure 3.* **Gradient Analysis on the First Skip-Link Block.**
We visualize the training dynamics during the early phase of the
joint training stage. *(a)* Gradient norms of the skip and main
branches, measured by $\mathbb{E}[\|\mathbf{g}\|^2]$, together with their MA(Moving
Average) counterparts, where the MA serves as an estimation to
$\|\mathbb{E}[\mathbf{g}]\|^2$. *(b)* Trace statistics computed as $\mathbb{E}[\|\mathbf{g}\|^2] - \|\mathbb{E}[\mathbf{g}]\|^2$,
characterizing the variance of the gradients over training. *(c)* The
cosine similarity between $g^{\mathrm{skip}}$ and $g^{\mathrm{main}}$ remains close to zero,
indicating approximate orthogonality. *(d)* Scatter plot of $\delta$ defined
in Assumption 4.2. Additional analysis is provided in Appendix C.

alyzing which layers to detach and how to configure fusion
strides are presented in Section 6.3.

An overview of the training pipeline is illustrated in Fig. 1.

### 3.3. Gradient Dynamics Analysis

To further justify the necessity of detachment, we analyze
gradient statistics during the early phase of joint training
(i.e., the first 1.3k steps). We validate Assumption 4.2 by
monitoring the gradient flow at the first skip-link block
(Block 6) during the the early phase of the joint training
stage. As illustrated in Fig. 3, the empirical results are
consistent with our hypothesis. The skip path is initially
dominated by high-variance noise rather than coherent sig-
nal, remains approximately orthogonal to the main path, and
exhibits negligible cross-covariance cancellation. Detailed
experimental settings and further analysis are provided in
Appendix C.3.

## 4. Theoretical Analysis

We provide a simplified analysis to build intuition for the
proposed detachment strategy. Our analysis focuses on
the *local* optimization behavior during the early stage of
joint training, with the goal of explaining why gradient
detachment leads to improved empirical stability.

### 4.1. Why skip-links help: a Bayes-risk decomposition

We first justify the forward benefit of fusing shallow skip
features $B = b(X)$ with the deep main-path representa-
tion $A = a(X)$. While deep networks are theoreti-
cally universal approximators, in practice, they act as in-
formation bottlenecks that may attenuate local details es-
sential for dense understanding. We quantify this benefit
by comparing the optimal population risk of a fusion pre-
dictor $f_{\mathrm{skip}}(X) = G([A; B])$ versus a main-only predictor
$f_{\mathrm{main}}(X) = F(A)$.

**Proposition 4.1** (Skip features provide complementary pre-
dictive information)**.** *Consider the squared loss $\mathcal{R}(f) \triangleq
\mathbb{E}(Y - f(X))^2$. The reduction in Bayes risk achievable by
incorporating the skip feature $B$ is exactly the conditional
variance explained by $B$ given $A$:*

$$\Delta\mathcal{R} \triangleq \inf_F \mathcal{R}\big(F(A)\big) - \inf_G \mathcal{R}\big(G([A;B])\big)$$
$$= \mathbb{E}[\mathrm{Var}(\mathbb{E}[Y \mid A, B] \mid A)] \qquad (2)$$
$$= \mathbb{E}\Big[\big(\mathbb{E}[Y \mid A, B] - \mathbb{E}[Y \mid A]\big)^2\Big] \geq 0.$$

*Moreover, $\Delta\mathcal{R} > 0$ whenever $B$ provides addi-
tional predictive information beyond $A$, i.e., whenever
$\mathbb{P}(\mathbb{E}[Y \mid A, B] \neq \mathbb{E}[Y \mid A]) > 0$ (equivalently, $Y \not\perp B \mid
A$).*

**Interpretation.** Shallow features can complement deep
representations. Eq. (2) formalizes a simple condition under
which fusing $B$ with $A$ strictly improves the optimal achiev-
able risk: the skip feature $B$ must contain task-relevant
information that is not already captured by the deep rep-
resentation $A$. In practice, deep encoders are shaped by
architectural and training biases (e.g., reduced spatial reso-
lution due to striding/pooling and a tendency to emphasize
coarse semantic abstractions), which can reduce sensitivity
to fine-grained local cues that are useful for dense ground-
ing. Thus, even when the model class is expressive, the
learned deep feature $A$ may not retain all predictive cues
needed for the downstream objective. Since $B$ is extracted
from earlier layers, it can preserve complementary localized
or high-frequency information and thereby reduce the irre-
ducible error captured by $\Delta\mathcal{R}$. (See Appendix B.1 for the
detailed derivation.)

### 4.2. Gradient Estimation: Pathwise Decomposition and Variance Structure

Consider the training objective $\mathcal{L}(\theta)$. In architectures with
skip-based fusion, the stochastic gradient on the shared
parameters at iteration $t$ admits a pathwise decomposition:

$$\mathbf{g}_t = \mathbf{g}_t^{\mathrm{main}} + \mathbf{g}_t^{\mathrm{skip}}, \qquad (3)$$

where $\mathbf{g}_t^{\mathrm{main}}$ is propagated through the main visual backbone,
and $\mathbf{g}_t^{\mathrm{skip}}$ is induced by the skip-fusion pathway.

**Empirical motivation.** In the early stage of joint optimization (e.g., the initial phase of FFT(Full Fine-Tuning) after warmup), we observe two recurring patterns (Fig. 3): (i) *gradient misalignment*: $\cos(\mathbf{g}^{\text{main}}, \mathbf{g}^{\text{skip}})$ is often near-zero or negative; (ii) *variance dominance*: the second moment (or a running variance proxy) of $\mathbf{g}^{\text{skip}}$ is substantially larger than that of $\mathbf{g}^{\text{main}}$.

**Mean–covariance decomposition.** Let $\mathbf{m} \triangleq \mathbb{E}[\mathbf{g}^{\text{main}}]$ and $\mathbf{s} \triangleq \mathbb{E}[\mathbf{g}^{\text{skip}}]$, and define $\Sigma_{\text{m}} \triangleq \text{Cov}(\mathbf{g}^{\text{main}})$, $\Sigma_{\text{s}} \triangleq \text{Cov}(\mathbf{g}^{\text{skip}})$, and $\Sigma_{\text{ms}} \triangleq \text{Cov}(\mathbf{g}^{\text{main}}, \mathbf{g}^{\text{skip}})$. Then the second moment of the full estimator $\mathbf{g}_{\text{full}} \triangleq \mathbf{g}^{\text{main}} + \mathbf{g}^{\text{skip}}$ can be written as

$$\mathbb{E}\big[\|\mathbf{g}_{\text{full}}\|^2\big] = \|\mathbf{m} + \mathbf{s}\|^2 + \text{tr}\big(\Sigma_{\text{m}} + \Sigma_{\text{s}} + \Sigma_{\text{ms}} + \Sigma_{\text{ms}}^\top\big), \tag{4}$$

where the trace terms summarize the total variance and cross-covariance contributions. (We include the short derivation in Appendix B.2 for completeness.)

**Assumption 4.2** (Early-phase pathwise gradient statistics). During the early stage of joint optimization, the skip-path gradient is *noise-dominant* and only weakly beneficial in expectation:

1. **Variance dominance:** $\text{tr}(\Sigma_{\text{s}}) \geq c \cdot \text{tr}(\Sigma_{\text{m}})$ for some $c \gg 1$.

2. **Weak (or adverse) mean alignment:** $\langle \mathbf{m}, \mathbf{s} \rangle \leq 0$ and $\|\mathbf{s}\| \leq \rho \|\mathbf{m}\|$ for a small $\rho$.

3. **Limited cross-covariance cancellation (mild):** $\big|\text{tr}(\Sigma_{\text{ms}} + \Sigma_{\text{ms}}^\top)\big| \leq \delta \cdot \text{tr}(\Sigma_{\text{s}})$ for some $\delta \in [0, 1)$.

**SNR and the role of detachment.** We measure the quality of a stochastic gradient estimator $\mathbf{g}$ via a directional signal-to-noise ratio (SNR),

$$\eta(\mathbf{g}) \triangleq \frac{\|\mathbb{E}[\mathbf{g}]\|^2}{\mathbb{E}[\|\mathbf{g}\|^2]} = \frac{\|\mathbb{E}[\mathbf{g}]\|^2}{\|\mathbb{E}[\mathbf{g}]\|^2 + \text{tr}(\text{Cov}(\mathbf{g}))}. \tag{5}$$

Eq. (4) shows that $\mathbf{g}_{\text{full}}$ inherits a potentially large variance term $\text{tr}(\Sigma_{\text{s}})$ from the skip path. Under Assumption 4.2, this variance dominates the denominator in (5), while the skip mean $\mathbf{s}$ provides little (or even adverse) contribution. Detaching the skip pathway on shared parameters corresponds to using $\mathbf{g}_{\text{detach}} \triangleq \mathbf{g}^{\text{main}}$, which removes this dominant source of stochastic variability and increases the effective directional SNR in the early phase.

**One-step progress under smoothness.** To connect the estimator-level view to optimization progress, assume $\mathcal{L}$ is $L$-smooth and consider the update $\theta^+ = \theta - \gamma \mathbf{g}$. A standard smoothness argument yields (see Appendix B.3)

$$\mathbb{E}\big[\mathcal{L}(\theta^+)\big] \leq \mathcal{L}(\theta) - \gamma \langle \nabla\mathcal{L}(\theta), \mathbb{E}[\mathbf{g}] \rangle + \frac{L\gamma^2}{2} \mathbb{E}\big[\|\mathbf{g}\|^2\big]. \tag{6}$$

**Proposition 4.3** (When detachment improves early-phase stability). *Let* $\mathbf{g}_{full} = \mathbf{g}^{main} + \mathbf{g}^{skip}$ *and* $\mathbf{g}_{detach} = \mathbf{g}^{main}$. *If*

$$\langle \nabla\mathcal{L}(\theta), \mathbf{s} \rangle \leq \frac{L\gamma}{2}\big(\mathbb{E}\|\mathbf{g}_{full}\|^2 - \mathbb{E}\|\mathbf{g}_{detach}\|^2\big), \tag{7}$$

*then* $\mathbb{E}[\mathcal{L}(\theta - \gamma\mathbf{g}_{detach})] \leq \mathbb{E}[\mathcal{L}(\theta - \gamma\mathbf{g}_{full})]$.

**Interpretation.** Condition (7) makes the bias–variance tradeoff explicit. The left-hand side measures the additional expected descent contributed by the skip mean gradient $\mathbf{s}$, while the right-hand side captures the extra second-moment penalty incurred when adding the skip component. In the early phase, Assumption 4.2 suggests that $\mathbf{s}$ is small and weakly aligned, whereas $\mathbb{E}\|\mathbf{g}_{\text{full}}\|^2 - \mathbb{E}\|\mathbf{g}_{\text{detach}}\|^2$ is dominated by the high-variance skip term, making detachment more stable and often yielding better expected one-step progress. This perspective is consistent with analyses of biased SGD (Ajalloeian & Stich, 2020), where a small early-phase bias can be beneficial if it substantially reduces effective gradient noise.

# 5. Empirical Analysis: $R$-probe

Hallucinations in OCR, such as misrecognizing "appie" as "apple", can originate from two distinct failure modes: (i) loss of fine-grained visual information during visual tokenization (Wei et al., 2024), or (ii) representational misalignment, where visual tokens are projected into a space that is poorly used by downstream language models(Huang et al., 2024). Metrics based solely on final textual outputs conflate these factors, making it difficult to distinguish visual encoding errors from language-side inference failures (He et al., 2025).

To disentangle these effects, we introduce the **Reconstruction Probe ($R$-Probe)**, a reconstruction-based diagnostic designed to evaluate whether visual tokens preserve sufficient information *and* are aligned with an LLM-style decoding regime. Crucially, $R$-Probe operationalizes reconstructability under a constrained, LLM-aligned decoder, using reconstruction performance as a proxy for visual fidelity and representational alignment, rather than as a general-purpose auto-encoding objective.

## 5.1. $R$-Probe as an LLM-Aligned Diagnostic Head

Figure 4 (right) illustrates the $R$-Probe architecture. The probe attaches a lightweight reconstruction head to a pretrained multimodal backbone, while strictly freezing the ViT encoder and the adapter. These frozen components constitute the subject of evaluation and define the vision-language bridging mechanism under analysis.

The probe itself consists of a shallow Transformer decoder followed by an MLP projector that maps decoder states

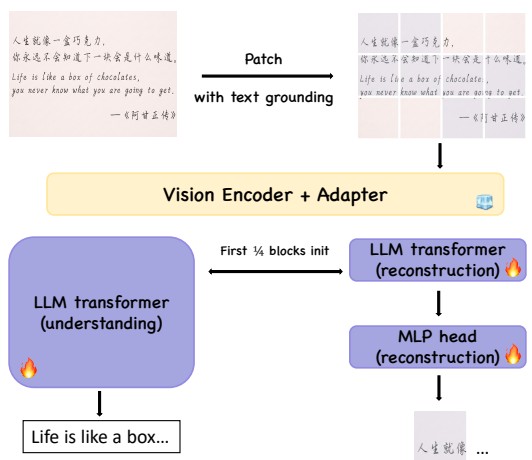

*Figure 4.* **Overview of the $R$-Probe and its use as an auxiliary training signal.** The $R$-Probe corresponds to the *right* reconstruction branch, which evaluates information retention and LLM-aligned consumability of visual tokens under frozen encoders. When attached to the full model (left), the same reconstruction head supplies an auxiliary loss during training, encouraging visually faithful representations without modifying the primary LLM decoding objective.

back to pixel space, as in Fig 4 right branch. The decoder is initialized from the first quarter of layers of a pre-trained language model(e.g LLaMA-3.1-8B). This choice intentionally restricts the probe's expressive capacity: early LLM layers operate in a relatively modality-agnostic regime, whereas deeper layers increasingly encode language-specific abstractions and priors (Liang et al., 2022). By restricting depth and freezing the backbone, successful reconstruction is only possible if visual tokens are both information-rich and projected into a subspace directly used by an LLM-style decoder.

### 5.2. Context-Aware Sequence Modeling

To reflect the conditional nature of OCR inference, we adopt a context-aware reconstruction protocol (Figure 5). Input images are rescaled to multiples of 448 and divided into non-overlapping $448 \times 448$ tiles. To match the LLM's input requirements, we employ a $2 \times 2$ pooling/merging strategy on the $14 \times 14$ ViT patch embeddings, condensing them into single visual tokens as per our adapter architecture.

Rather than reconstructing images in isolation, we construct the input sequence as

$$\mathcal{S} = [\mathbf{E}_{\text{context\_img}}, \mathbf{E}_{\text{text}}, \mathbf{E}_{\text{target\_img}}], \tag{8}$$

where $\mathbf{E}_{\text{target\_img}}$ corresponds to the image region containing the text of interest. Global 2D RoPE is applied prior to reordering, ensuring that absolute spatial relationships are preserved across visual tokens. This formulation enforces conditional reconstruction: the probe must recover the target region while attending to both surrounding visual context

and textual prompts, rather than performing unconditional image reconstruction. We further analyze $R$-Probe behavior under missing textual or visual inputs in the Appendix D.2, to isolate the respective roles of language context and visual evidence.

### 5.3. Diagnostic Rationale: Why Reconstruction Matters

The diagnostic value of $R$-Probe relies on the premise that effective visual tokens must be compatible with the LLM's initialization space to support reconstruction. Importantly, $R$-Probe is not a pixel-level autoencoder; it serves as a *semantic consistency check* between visual and language representations. We assess its validity and sensitivity through the following analyses.

**Semantic Dependency Verification.** We examine whether reconstruction relies on the vision–language interface rather than visual input alone. As shown by the modality ablation study (Appendix D.2), reconstruction quality degrades substantially when visual regions are masked. However, providing textual descriptions reduces the reconstruction loss from 1.980 to 1.103 (MSE), despite the absence of visual signals. This result indicates that the decoder exploits semantic information from language tokens to guide reconstruction, confirming that $R$-Probe measures cross-modal alignment.

**Sensitivity to Feature Quality.** We examine whether $R$-Probe reflects differences in visual representation quality. We evaluate optimization efficiency by the number of steps to reach MSE $< 0.75$ and report the final reconstruction loss, where 0.75 is an empirically chosen threshold indicating good reconstruction quality. As shown in Appendix D.3, detached multi-layer aggregation reaches the target loss faster (2158 $\rightarrow$ 1689 steps) and achieves a lower final error (0.698 vs. 0.724) than the baseline, indicating that $R$-Probe is sensitive to feature quality.

**Robustness and Correlation with Downstream Performance.** We further test whether this sensitivity is consistent across architectures and aligned with downstream tasks. Across four MLLM backbones, the same configuration consistently reduces optimization steps and final reconstruction loss (Table 7), demonstrating robust and architecture-agnostic behavior. Overall, reconstruction loss rankings induced by $R$-Probe show a clear association with downstream performance within each backbone, (Appendix D.4), supporting its use as a predictive diagnostic for comparing model configurations under a fixed backbone.

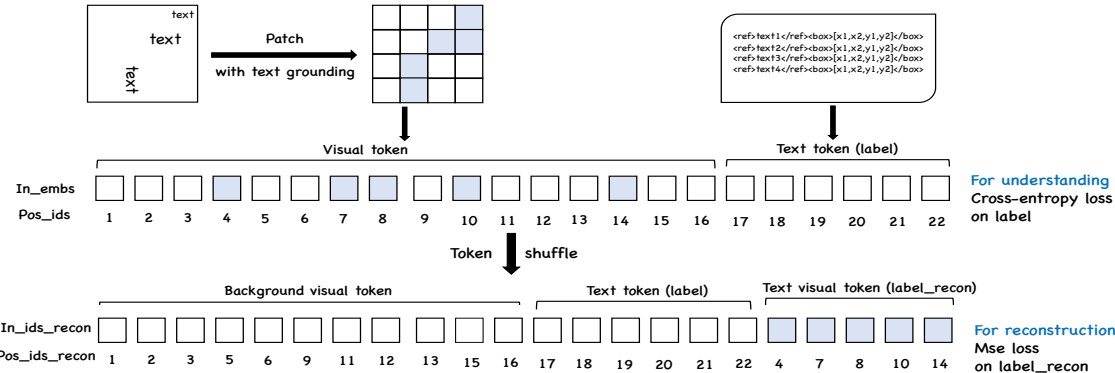

*Figure 5.* **Sequence Construction Strategy.** For reconstruction used in $R$-probe, we prioritize context (background image and text prompts) before the target image tokens. This forces the probe to verify if the adapter successfully projects visual features into a space that the LLM can utilize for context-dependent reconstruction.

## 6. Experiments: large-scale

In this section, we evaluate Detached Skip-Links at scale. We first describe the experimental setup and benchmark protocol, and then present ablation studies, comparisons with state-of-the-art fusion methods, and evaluations across different ViT backbones.

### 6.1. Experimental Setup

We adopt a two-stage training pipeline: (i) adapter pre-training (warm-up on the adapter only, then FFT) and (ii) supervised fine-tuning (SFT). Unless otherwise specified, we use LLaMA-3.1-8B as the base LLM and a 300M–400M parameter Vision Transformer as the visual encoder. To evaluate scalability, we pre-train on 5M multimodal samples and further fine-tune on 2M task-specific samples. Full details on data sources, task composition, and hyperparameters are provided in Appendix E.

As illustrated in Fig. 1, during adapter pre-training we freeze both the ViT encoder and the LLM, and optimize only the adapter. During FFT and SFT, we fine-tune the full model for OCR- and VQA-centric tasks using the same training recipe, with stage-specific learning rates and sequence lengths reported in Appendix E.3.

### 6.2. Benchmark Evaluation

We evaluate on 22 benchmarks spanning four categories: STEM Puzzle, General, Alignment, and OCR, covering a broad range of multimodal reasoning and perception tasks. For compactness, the main text reports the average score of each group, while per-benchmark results are deferred to Appendix G. We use a customized VLMEvalKit pipeline; implementation details are included in Appendix F.

### 6.3. Ablation Studies

**Detachment configuration.** In this section, we study how fusion density and gradient-flow control affect multi-layer visual feature fusion. We introduce two key hyperparameters: the sampling **stride** ($S$), which controls how densely intermediate ViT layers are selected, and the number of **detached layers** ($D$). Specifically, we extract feature maps every $S$ blocks from shallow to deep. Let $\mathcal{L} = \{\ell_1, \ell_2, \ldots, \ell_K\}$ denote the selected blocks ordered by depth, where $\ell_1$ is the shallowest. We apply stop-gradient to the shallowest $D$ layers $\{\ell_1, \ldots, \ell_D\}$, preventing gradients from the LLM objective from updating these layers through the skip-fusion path.

Figure 6 visualizes the ablation results, where bubble size and color indicate performance relative to the baseline (yellow). Two clear findings emerge. **(i) An intermediate fusion density is optimal:** denser fusion with smaller strides (e.g., $S = 3$ or 4) consistently outperforms sparse fusion (e.g., $S = 12$), highlighting the benefit of incorporating multi-level visual features. **(ii) Shallow-only detachment is robust:** Detaching shallow layers while keeping deeper layers trainable yields strong performance across fusion settings, whereas detaching deeper layers leads to instability. This suggests that our method is robust to detachment hyperparameters.

**Training with $R$-Probe as an Auxiliary Loss** We optionally employ the $R$-Probe as a self-supervised auxiliary objective to impose a structural consistency constraint on visual tokens (Figure 4). While this enhances OCR performance by preserving fine-grained details, it introduces slight trade-offs in abstract reasoning. We attribute this to distributional bias from the OCR-centric auxiliary data (see Appendix D.1).

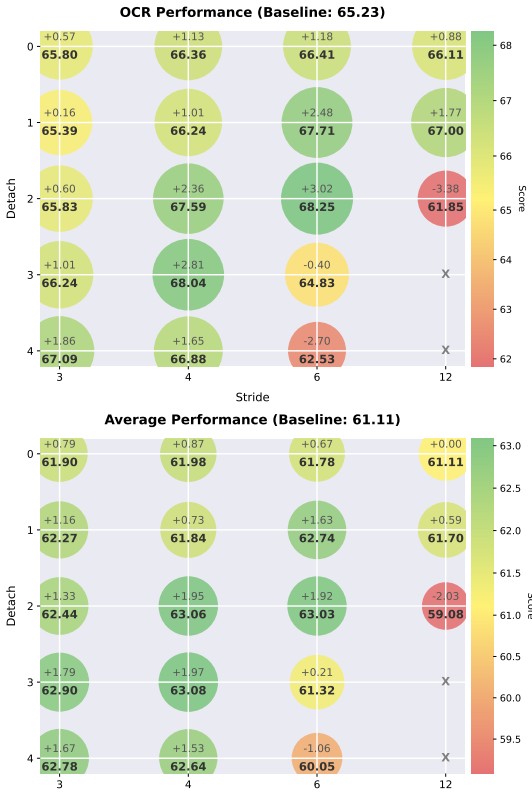

*Figure 6.* **Ablation study on feature sampling stride ($S$) and the number of detached layers ($D$).** The chart visualizes the OCR performance (top) and the Average score across all benchmarks (bottom).Green nodes indicate improvement, while red nodes indicate degradation.

## 6.4. Comparison with State-of-the-Art Methods

We compare our method with three representative multi-layer visual feature fusion methods: Dense Connector for MLLMs (DC) (Yao et al., 2024), Deepstack (Meng et al., 2024), and Multi-Layer Visual Feature Fusion (ML) (Lin et al., 2025). For fair comparison, all methods are trained from the same initialization on the same dataset, using identical settings. As shown in Table 1, our method achieves the strongest overall performance under matched training budgets. Following the official best settings of each baseline, we apply the detach operation to selected layers, which consistently improves performance and demonstrates its generalization. Specifically, we use DC with layer groups [1–12] and [12–23], DeepStack with Starting Layers=4, Interval=2, and N-layers=4, and ML with External Direct Fusion, applying detach to DC's [1–12] group and ML's last layer.

## 6.5. Experiments Across Different ViTs

To evaluate generalizability, we test Detached Skip-Links across diverse Vision Transformer backbones with differ-

*Table 1.* Performance Comparison (Categorized Averages). Full results in Table 9

| Setting | STEM | General | Align. | OCR | Overall |
|---|---|---|---|---|---|
| PE-baseline | 63.0 | 53.2 | 72.6 | 65.2 | 61.1 |
| Ours (PE-best) | 64.1 | **54.6** | **73.6** | **68.3** | **63.0** |
| DC | 63.2 | 54.0 | 72.5 | 66.7 | 62.0 |
| DC-detached | **64.2** | 54.4 | 72.8 | 67.6 | 62.6 |
| ML | 63.5 | 54.1 | 72.6 | 66.9 | 62.1 |
| ML-detached | 63.1 | 54.0 | 73.2 | 68.1 | 62.5 |
| DeepStack | 63.8 | 54.5 | 73.2 | 67.6 | 62.6 |

ent architectures and pre-training objectives. Specifically, beyond the Perception Encoder used in ablations, we evaluate InternViT-300M (448px), AimV2-L (patch14-224), and SigLip2-So400M (patch14-384). For each ViT, we compare the baseline w/o our proposed Detached Skip-Links method. Experimental results (categorized averages) are illustrated in Table 2. Consistent performance improvements across all tested ViTs demonstrate that our method possesses broad adaptability to different visual encoding architectures.

*Table 2.* Performance Across Different ViT Backbones (Categorized Averages). Full results in Table 10

| Setting | STEM | General | Align. | OCR | Overall |
|---|---|---|---|---|---|
| PE-baseline | 63.0 | 53.2 | 72.6 | 65.2 | 61.1 |
| PE-ours | 64.1 | 54.6 | 73.6 | 68.3 | 63.0 |
| $\Delta$ | **+1.1** | **+1.4** | **+1.0** | **+3.1** | **+1.9** |
| InternViT-baseline | 58.7 | 49.0 | 64.0 | 60.3 | 56.2 |
| InternViT-ours | 61.1 | 50.8 | 71.4 | 62.2 | 58.7 |
| $\Delta$ | **+2.4** | **+1.8** | **+7.4** | **+1.9** | **+2.5** |
| AimV2$_L$-baseline | 60.5 | 51.0 | 69.5 | 63.0 | 58.8 |
| AimV2$_L$-ours | 62.2 | 52.7 | 71.5 | 64.8 | 60.6 |
| $\Delta$ | **+1.7** | **+1.7** | **+2.0** | **+1.8** | **+1.8** |
| SigLip2$_{so400}$-baseline | 57.1 | 47.4 | 69.6 | 58.1 | 55.0 |
| SigLip2$_{so400}$-ours | 59.7 | 49.8 | 69.4 | 60.6 | 57.3 |
| $\Delta$ | **+2.6** | **+2.4** | **-0.2** | **+2.5** | **+2.3** |

## 7. Conclusions

We study an optimization challenge in OCR-centric ViT–LLM fusion, where shallow visual features are underutilized or distorted during joint training. To address this issue, we introduce a detached gradient strategy for multi-layer fusion, together with *R-Probe*, a reconstruction-based diagnostic for assessing fine-grained visual information preservation and LLM-aligned consumability.

Across large-scale experiments, our approach consistently improves OCR-centric performance while maintaining or improving general multimodal capabilities, and demon-

strates robust transfer across diverse ViT backbones. Beyond empirical gains, our analysis highlights the importance of decoupling feature aggregation from gradient propagation when integrating heterogeneous visual representations. We hope these findings provide practical guidance for designing and diagnosing ViT–LLM bridging mechanisms, particularly in document-level and fine-grained multimodal understanding settings.

## Acknowledgements

This paper is partially supported by the National Key Research and Development Program of China with Grant No. 2023YFC3341203 as well as the National Natural Science Foundation of China with Grant Number 62306014. This work was also supported by computational resources and data from Baidu AI Cloud. We are additionally grateful to Prof. Xue Tianfan for his helpful suggestions on the manuscript.

## Impact Statement

This work focuses on improving the reliability of OCR-centric multimodal models through better optimization and diagnostic tools. All training and evaluation data are desensitized and sourced from publicly available or synthetic datasets, and do not involve personally identifiable information. While our methods may benefit document understanding applications, including large-scale information processing systems, they do not introduce new data collection mechanisms or user-facing decision-making components. We do not foresee significant negative societal impacts arising directly from this work.

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

## A. Attention Visualization Details

We visualize shallow-layer attention maps following the protocol of (Darcet et al., 2024). Given an input image, we forward it through the vision encoder and extract the self-attention weights from a shallow transformer block. Here, we use the 4th transformer block as a representative shallow layer. We use the [CLS] token attention to patch tokens: we average attention weights over all heads and take the row corresponding to the [CLS] query, excluding the [CLS]→[CLS] entry. The resulting patch-level importance scores are reshaped into a 2D grid according to the ViT patch layout.

For visualization, we apply min–max normalization to map the scores to $[0, 1]$ and upsample the grid to the input image resolution using nearest-neighbor interpolation, producing a discrete patch-aligned heatmap. Unless otherwise specified, we enable the [CLS]-based attention mode by default. We repeat the same procedure for three checkpoints and concatenate the resulting heatmaps with the original image into a single panel for side-by-side comparison (original ‖ ori | baseline | detached), using a white background and fixed spacing.

## B. Additional Proofs for Section 4

### B.1. Proof of Proposition 4.1

**Proof.** We analyze the gap between the optimal risks of the main-only path and the fused path. Recall that under the squared loss $\mathcal{R}(f) = \mathbb{E}[(Y - f(X))^2]$, the Bayes optimal predictor is the conditional expectation of the target given the inputs. Therefore, the minimum achievable risks for the two scenarios are:

$$\mathcal{R}^*_{\text{main}} = \inf_F \mathcal{R}(F(A)) = \mathbb{E}\big[(Y - \mathbb{E}[Y \mid A])^2\big],$$
$$\mathcal{R}^*_{\text{fuse}} = \inf_G \mathcal{R}(G([A; B])) = \mathbb{E}\big[(Y - \mathbb{E}[Y \mid A, B])^2\big].$$

We can decompose the risk of the main branch by adding and subtracting the term $\mathbb{E}[Y \mid A, B]$ inside the quadratic expectation:

$$\mathcal{R}^*_{\text{main}} = \mathbb{E}\Big[\big((Y - \mathbb{E}[Y \mid A, B]) + (\mathbb{E}[Y \mid A, B] - \mathbb{E}[Y \mid A])\big)^2\Big]$$
$$= \underbrace{\mathbb{E}\big[(Y - \mathbb{E}[Y \mid A, B])^2\big]}_{=\mathcal{R}^*_{\text{fuse}}} + \underbrace{\mathbb{E}\big[(\mathbb{E}[Y \mid A, B] - \mathbb{E}[Y \mid A])^2\big]}_{\text{Gap term}}$$
$$+ 2\underbrace{\mathbb{E}\Big[(Y - \mathbb{E}[Y \mid A, B]) \cdot (\mathbb{E}[Y \mid A, B] - \mathbb{E}[Y \mid A])\Big]}_{\text{Cross term}}. \tag{9}$$

Now, we show that the cross term vanishes. By the Law of Iterated Expectations, conditioning on $A, B$ inside the expectation:

$$\text{Cross term} = \mathbb{E}_{A,B}\Big[\mathbb{E}\big[Y - \mathbb{E}[Y \mid A, B] \mid A, B\big] \cdot (\mathbb{E}[Y \mid A, B] - \mathbb{E}[Y \mid A])\Big]$$
$$= \mathbb{E}_{A,B}\Big[\underbrace{(\mathbb{E}[Y \mid A, B] - \mathbb{E}[Y \mid A, B])}_{0} \cdot (\ldots)\Big] = 0.$$

Substituting this back into Eq. (9), we obtain the risk difference:

$$\mathcal{R}^*_{\text{main}} - \mathcal{R}^*_{\text{fuse}} = \mathbb{E}\Big[(\mathbb{E}[Y \mid A, B] - \mathbb{E}[Y \mid A])^2\Big].$$

Using the definition of conditional variance $\text{Var}(Z \mid A) = \mathbb{E}[Z^2 \mid A] - (\mathbb{E}[Z \mid A])^2$, and setting $Z = \mathbb{E}[Y \mid A, B]$, we observe that:

$$\mathbb{E}[Z \mid A] = \mathbb{E}[\mathbb{E}[Y \mid A, B] \mid A] = \mathbb{E}[Y \mid A].$$

Thus, the gap term is exactly the expected conditional variance of the predictor:

$$\mathbb{E}\Big[(\mathbb{E}[Y \mid A, B] - \mathbb{E}[Y \mid A])^2\Big] = \mathbb{E}_A\Big[\text{Var}\big(\mathbb{E}[Y \mid A, B] \mid A\big)\Big].$$

This confirms Eq. (2). Since the squared term is always non-negative, $\mathcal{R}^*_{\text{main}} \geq \mathcal{R}^*_{\text{fuse}}$.

**Condition for Equality.** The gap is zero if and only if $\mathbb{E}[Y \mid A, B] = \mathbb{E}[Y \mid A]$ almost surely. This occurs when $Y$ is conditionally independent of $B$ given $A$ (i.e., $Y \perp B \mid A$). In our context, this would imply that the deep feature $A$ has preserved all information from $B$ relevant to $Y$, meaning no information bottleneck exists. Conversely, if the main path attenuates relevant information (as discussed in Section 4.1), strict inequality holds. □

## B.2. Derivation of Eq. (4)

Let $\mathbf{g}_{\text{full}} = \mathbf{g}^{\text{main}} + \mathbf{g}^{\text{skip}}$ with means $\mathbf{m} = \mathbb{E}[\mathbf{g}^{\text{main}}]$, $\mathbf{s} = \mathbb{E}[\mathbf{g}^{\text{skip}}]$. Write $\mathbf{g}^{\text{main}} = \mathbf{m} + \boldsymbol{\epsilon}_{\text{m}}$ and $\mathbf{g}^{\text{skip}} = \mathbf{s} + \boldsymbol{\epsilon}_{\text{s}}$, where $\mathbb{E}[\boldsymbol{\epsilon}_{\text{m}}] = \mathbb{E}[\boldsymbol{\epsilon}_{\text{s}}] = 0$. Then

$$\mathbb{E}\|\mathbf{g}_{\text{full}}\|^2 = \|\mathbf{m} + \mathbf{s}\|^2 + \mathbb{E}\|\boldsymbol{\epsilon}_{\text{m}} + \boldsymbol{\epsilon}_{\text{s}}\|^2 = \|\mathbf{m} + \mathbf{s}\|^2 + \mathbb{E}\|\boldsymbol{\epsilon}_{\text{m}}\|^2 + \mathbb{E}\|\boldsymbol{\epsilon}_{\text{s}}\|^2 + 2\mathbb{E}\langle\boldsymbol{\epsilon}_{\text{m}}, \boldsymbol{\epsilon}_{\text{s}}\rangle.$$

Using $\mathbb{E}\|\boldsymbol{\epsilon}\|^2 = \operatorname{tr}(\operatorname{Cov}(\cdot))$ and $2\mathbb{E}\langle\boldsymbol{\epsilon}_{\text{m}}, \boldsymbol{\epsilon}_{\text{s}}\rangle = \operatorname{tr}(\Sigma_{\text{ms}} + \Sigma_{\text{ms}}^\top)$ yields Eq. (4). □

## B.3. One-step smoothness bound

Assume $\mathcal{L}$ is $L$-smooth, i.e., $\mathcal{L}(\theta') \leq \mathcal{L}(\theta) + \langle\nabla\mathcal{L}(\theta), \theta' - \theta\rangle + \frac{L}{2}\|\theta' - \theta\|^2$. Substitute $\theta' = \theta - \gamma\mathbf{g}$ and take expectation to obtain Eq. (6). □

## B.4. Proof of Proposition 4.3

Apply Eq. (6) to $\mathbf{g}_{\text{full}}$ and $\mathbf{g}_{\text{detach}}$ and subtract:

$$\mathbb{E}\mathcal{L}(\theta - \gamma\mathbf{g}_{\text{full}}) - \mathbb{E}\mathcal{L}(\theta - \gamma\mathbf{g}_{\text{detach}}) \leq -\gamma\langle\nabla\mathcal{L}(\theta), \mathbf{s}\rangle + \frac{L\gamma^2}{2}\left(\mathbb{E}\|\mathbf{g}_{\text{full}}\|^2 - \mathbb{E}\|\mathbf{g}_{\text{detach}}\|^2\right),$$

where $\mathbf{s} = \mathbb{E}[\mathbf{g}^{\text{skip}}]$. Rearranging gives the sufficient condition (7) for detachment to yield no worse expected one-step loss. □

# C. Details on Gradient Statistics Measurement

This section details the methodology used to separate and analyze the skip-path and main-path gradients presented in Figure 3.

## C.1. Online Measurement Strategy

In standard backpropagation, gradients from the skip and main branches are aggregated automatically. To decouple them for analysis without disrupting the training graph, we employed a *Double Backward* strategy with random state preservation:

1. **State Checkpointing:** We save the current state of the Random Number Generator (RNG) to ensure consistent dropout masks and stochastic operations.

2. **Main-Path Isolation:** We perform a forward pass where the skip connection is detached ($y = \mathcal{F}(x) + x.\text{detach}()$). The subsequent backward pass yields $g^{\text{main}} = \nabla_\theta\mathcal{L}_{\text{detach}}$.

3. **State Restoration:** The RNG state is restored.

4. **Full Gradient Computation:** A standard forward and backward pass is executed to obtain the total gradient $g^{\text{full}}$.

5. **Decomposition:** The skip-path gradient is derived via subtraction: $g^{\text{skip}} = g^{\text{full}} - g^{\text{main}}$.

## C.2. Metric Definitions

To empirically verify the statistical assumptions in Section 4, we compute the following window-based gradient statistics.

**Signal–Noise Decomposition.** For a random gradient vector $g$, its second moment admits the standard decomposition

$$\mathbb{E}\left[\|g\|^2\right] = \|\mathbb{E}[g]\|^2 + \operatorname{tr}(\operatorname{Var}(g)). \tag{10}$$

In practice, expectations are approximated using a sliding window of length $K$. Specifically, given gradients $\{g_{t-K+1}, \ldots, g_t\}$, we define the window mean

$$\hat{g}_t = \frac{1}{K} \sum_{i=1}^{K} g_{t-K+i}, \tag{11}$$

and the corresponding empirical variance trace

$$\text{tr}(\hat{\Sigma}_t) = \frac{1}{K} \sum_{i=1}^{K} \|g_{t-K+i}\|^2 - \|\hat{g}_t\|^2. \tag{12}$$

Accordingly, the instantaneous squared norm $\|g_t\|^2$ serves as a proxy for the total gradient energy, while $\|\hat{g}_t\|^2$ approximates the signal component. The pronounced gap between these quantities in Figure 3a indicates that gradient variance dominates the optimization dynamics in the early training stage.

**Cross-Covariance Strength ($\delta$).** To quantify the relative magnitude of the cross-covariance term between the main and skip branches, we consider windowed gradients $\{g_i^{\text{main}}, g_i^{\text{skip}}\}_{i=1}^K$. Let

$$\hat{m}_t = \frac{1}{K} \sum_{i=1}^{K} g_{t-K+i}^{\text{main}}, \qquad \hat{s}_t = \frac{1}{K} \sum_{i=1}^{K} g_{t-K+i}^{\text{skip}},$$

denote their respective window means. We estimate the trace of the symmetric cross-covariance as

$$\left| \text{tr}(\Sigma_{ms} + \Sigma_{ms}^\top) \right| \approx 2 \left| \frac{1}{K} \sum_{i=1}^{K} \langle g_{t-K+i}^{\text{main}}, g_{t-K+i}^{\text{skip}} \rangle - \langle \hat{m}_t, \hat{s}_t \rangle \right|. \tag{13}$$

Similarly, the variance trace of the skip branch is estimated by

$$\text{tr}(\Sigma_s) \approx \frac{1}{K} \sum_{i=1}^{K} \|g_{t-K+i}^{\text{skip}}\|^2 - \|\hat{s}_t\|^2. \tag{14}$$

We then define the empirical cross-covariance ratio

$$\delta_t = \frac{\left| \text{tr}(\Sigma_{ms} + \Sigma_{ms}^\top) \right|}{\text{tr}(\Sigma_s) + \epsilon}, \qquad \epsilon = 10^{-12}, \tag{15}$$

which serves as a practical proxy for the constant $\delta$ in Assumption 4.2. As shown in Figure 3d, $\delta_t$ remains small throughout training, indicating that the cross-covariance is negligible relative to the variance of the skip-path gradients.

### C.3. Detailed Empirical Analysis of Gradient Statistics

In Section 3.3, we summarized the gradient dynamics of the first skip-link block. Here, we provide a detailed interpretation of the empirical observations shown in Figure 3.

- **Variance Dominance and Phase Transition (Fig. 3a):** During the early stage of training (approximately the first 300 steps), the gradient norm of the skip branch, $\|g^{\text{skip}}\|$, is consistently larger than that of the main branch, $\|g^{\text{main}}\|$. More importantly, the instantaneous skip-path norm significantly exceeds the squared norm of its short-horizon moving average, i.e., $\|g^{\text{skip}}\|^2 \gg \|\hat{g}^{\text{skip}}\|^2$, indicating that the skip-path gradients are dominated by high-variance fluctuations rather than a coherent mean signal. As training progresses, we observe a clear regime transition in which $\|g^{\text{main}}\|$ surpasses $\|g^{\text{skip}}\|$, marking the shift from initialization-driven dynamics to effective feature learning.

- **Approximate Orthogonality (Fig. 3c):** The cosine similarity between $g^{\text{skip}}$ and $g^{\text{main}}$ fluctuates tightly around zero throughout training. This behavior empirically supports the weak mean-alignment assumption, suggesting that the stochastic noise introduced by the skip branch is approximately orthogonal to the effective gradient direction of the main branch.

- **Negligible Cross-Covariance Cancellation (Fig. 3d):** The empirical cross-covariance ratio $\delta_t$ remains consistently small (typically below 0.1) over the entire training trajectory. This observation indicates that the interaction between the main and skip branches is insufficient to offset the variance of the skip-path gradients. Consequently, the cross-covariance term plays a negligible role in practice, justifying its omission in our theoretical analysis.

## C.4. Robustness to learning rate.

| LR | $t_{\text{trans}}$ | median(cos) | median($\delta$) |
|------|------|----------|----------|
| 8e-5 | 400 | -0.0011 | 0.0577 |
| 2e-5 | 900 | -0.0005 | 0.0624 |
| 5e-6 | / | 0.0008 | 0.0315 |

*Table 3.* Learning-rate robustness summary of early-phase gradient statistics

We further evaluate the robustness of our gradient-statistics analysis under different learning rates, while keeping the model architecture and data pipeline fixed. Unless otherwise specified, all main experiments are conducted with a learning rate of $1 \times 10^{-5}$, with global batch size of 128, using Adam as optimizer.

Across a wide range of learning rates, we observe qualitatively consistent behaviors during the early phase of training: (i) *variance dominance* in the skip-path gradients, evidenced by the instantaneous norm $\|g_t^{\text{skip}}\|$ being substantially larger than the squared norm of its short-horizon window mean, and typically comparable to or larger than $\|g_t^{\text{main}}\|$; (ii) near-orthogonality between the main and skip branches, with cosine similarity fluctuating tightly around zero; and (iii) weak cross-covariance cancellation, with the empirical ratio $\delta_t$ remaining small (e.g., below 0.1 in our measurements).

The primary effect of changing the learning rate is a rescaling of the time axis, manifested as a shift in the step at which the training dynamics transition between regimes. Using a reproducible definition of the transition step $t_{\text{trans}}$—defined as the first step after which a running-window median of $\|g^{\text{skip}}\|/\|g^{\text{main}}\|$ remains below 1 for consecutive windows—we find that larger learning rates generally induce earlier transitions, while smaller learning rates delay this transition (Table 3).

At sufficiently large learning rates, we observe a qualitative inversion of the relative gradient magnitudes, where $\|g^{\text{main}}\|$ can dominate $\|g^{\text{skip}}\|$ even in the initial training steps. Notably, this inversion does not contradict our analysis, as Assumption 4.2 is intended as a local characterization of the early optimization regime under standard training settings, rather than a global statement valid for arbitrarily large step sizes.

# D. R-probe Results

## D.1. Training with R-Probe as Auxiliary Loss

To investigate the cross-modal interaction between the R-probe reconstruction mechanism and Optical Character Recognition (OCR) capabilities, we conducted a specialized ablation study. We constructed a dataset with bounding box annotations for text regions across various domains, including scene text, documents, and charts. By training the R-probe jointly with Multimodal Large Language Model (MLLM) OCR tasks, we aimed to determine if visual reconstruction objectives could synergize with text recognition objectives.

As shown in Table 4, we observe a mutual enhancement between the two tasks, particularly for benchmarks that rely heavily on pure visual perception and text recognition. For instance, OCRBench scores improved from 714.0 to 721.0, and DocVQA improved from 71.4 to 72.1. However, for tasks requiring complex reasoning over visual elements (e.g., CharXiv_RQ), we observed a slight performance trade-off, where scores decreased from 43.9 to 42.8. This suggests that while R-probe significantly strengthens low-level visual grounding and recognition features, it may introduce a slight interference in high-level semantic reasoning pathways when trained conjointly.

*Table 4.* Comparison of OCR capabilities with and without R-probe. "Baseline+R-probe" indicates the model jointly trained with the R-probe reconstruction objective. Note the improvement in pure OCR tasks (OCRBench, DocVQA) versus the trade-off in reasoning-heavy tasks (CharXiv_RQ).

| Method | OCRBench | DocVQA | TextVQA | ChartQA | OCRVQA | AI2D | CharXiv_DQ | CharXiv_RQ |
|--------|----------|--------|---------|---------|--------|------|------------|------------|
| Baseline | 714.0 | 71.4 | 68.7 | 70.0 | 55.0 | 72.2 | 69.7 | 43.9 |
| Baseline + R-probe | **721.0** | **72.1** | **68.9** | **70.1** | **55.4** | 69.2 | 68.3 | 42.8 |

## D.2. Ablation Study of R-probe Modalities

To quantify the information contribution of language tokens versus image tokens in the reconstruction process, we designed a set of ablation experiments involving masking image regions and removing language tokens (Table 5). In the "Masked" setting, the bounding box regions in the input image are replaced with black pixels ($R = G = B = 0$). In the "w/o Lang" setting, all language tokens are replaced with a special `[UNK]` token.

The results demonstrate that the presence of language tokens significantly aids visual reconstruction. Comparing Setting 1 (Masked, w/o Lang) and Setting 2 (Masked, w/ Lang), the reconstruction loss drops from 1.980 to 1.103, indicating that the language description provides crucial cues for reconstructing the missing visual information. This provides evidence that the R-probe Transformer possesses a degree of semantic understanding, effectively bridging the modality gap between text and image.

*Table 5.* Ablation study on input modalities for the R-probe. Lower Final Loss indicates better reconstruction quality.

| ID | Setting | Final Loss (MSE) |
|---|---|---|
| 1 | Masked Image, w/o Language | 1.980 |
| 2 | Masked Image, w/ Language | 1.103 |
| 3 | Full Image, w/o Language | 0.757 |
| 4 | Full Image, w/ Language | **0.724** |

## D.3. Convergence Analysis and Architecture Configuration

We further analyzed the convergence behavior and reconstruction quality of the R-probe under different layer configurations and backbone architectures. Convergence is defined as the number of steps required for the per-token MSE loss to drop below 0.75, representing a visually coherent reconstruction threshold.

Table 6 presents the impact of feature selection from different ViT layers. The "Detached" setting, where gradients are not backpropagated to the main ViT backbone, combined with multi-layer feature aggregation, yields the fastest convergence (1689 steps) and the lowest final loss (0.698). This suggests that aggregating hierarchical features from multiple depths provides a richer representation for reconstruction than using only superficial or deep layers alone.

*Table 6.* Impact of layer selection and gradient detachment on R-probe convergence and performance. "Steps" denotes steps to reach MSE < 0.75.

| ID | Configuration | Steps ($<$0.75) | Final Loss |
|---|---|---|---|
| 1 | Baseline | 2158 | 0.724 |
| 2 | Ours Stride=12 (w/o Detach) | 1935 | 0.719 |
| 3 | Ours Stride=12 (Detached) | 1956 | 0.716 |
| 4 | Ours Stride=6 (w/o Detach) | 1834 | 0.709 |
| 5 | Ours Stride=6 (Detached) | **1689** | **0.698** |

## D.4. Correlation Between R-Probe Reconstruction Loss and Downstream Performance

As discussed in the main text, $R$-Probe is designed not only to provide stable measurements across architectures, but also to reflect downstream perceptual performance. To support this claim, we analyze the relationship between R-Probe reconstruction loss and downstream benchmark results.

We first report the robustness of the optimized R-Probe configuration across different MLLM backbones (AimV2, InternViT, SigLip2). As shown in Table 7, the detached multi-layer configuration consistently reduces both the required optimization steps and the final reconstruction loss across all tested architectures, indicating architecture-agnostic behavior.

Building on the robustness results, we examine whether reconstruction loss aligns with downstream task performance under a fixed backbone.

Within each backbone, configurations with lower reconstruction loss consistently achieve higher OCR and general benchmark

*Table 7.* Performance of R-probe across different MLLM backbones. "Ours" refers to the best detached multi-layer configuration (Setting ID=5 from Table 6).

| Backbone | Setting | Steps ($<$0.75) | Final Loss |
|---|---|---|---|
| PE | Original | 2158 | 0.724 |
| | **Ours** | **1689** | **0.698** |
| AimV2 | Original | 2301 | 0.741 |
| | **Ours** | **1889** | **0.706** |
| InternViT | Original | 2189 | 0.735 |
| | **Ours** | **1835** | **0.703** |
| SigLip2 | Original | 2535 | 0.748 |
| | **Ours** | **2089** | **0.721** |

scores. This consistent within-backbone ranking indicates that $R$-Probe serves as a predictive diagnostic for comparing perceptual quality across model variants sharing the same architecture.

*Table 8.* Relationship between R-Probe reconstruction loss and downstream benchmark performance within each backbone.

| Backbone | Setting | Recon. Loss $\downarrow$ | OCR Score $\uparrow$ | General Score $\uparrow$ |
|---|---|---|---|---|
| PE | Original | 0.724 | 65.2 | 61.1 |
| | Ours Stride=12 (w/o Detach) | 0.719 | 66.1 | 61.1 |
| | Ours Stride=12 (Detached) | 0.716 | 67.0 | 61.7 |
| | Ours Stride=6 (w/o Detach) | 0.709 | 66.4 | 61.8 |
| | Ours Stride=6 (Detached) | 0.698 | 68.3 | 63.0 |
| AimV2 | Original | 0.741 | 63.0 | 58.8 |
| | Ours | 0.706 | 64.8 | 60.6 |
| InternViT | Original | 0.735 | 60.3 | 56.2 |
| | Ours | 0.703 | 62.2 | 58.7 |
| SigLip2 | Original | 0.748 | 58.1 | 55.0 |
| | Ours | 0.721 | 60.6 | 57.3 |

# E. Training Details

## E.1. Overview

For all model configurations, we adopt a two-stage training paradigm consisting of (i) adapter pre-training and (ii) supervised fine-tuning (SFT). The training data is sourced from a mixture of internal collections and publicly available datasets.[1] Unless otherwise specified, we use `LLaMA3.1-8B` as the base language model and a ViT visual encoder with 300M–400M parameters.

We sample 5M multimodal examples for adapter pre-training and 2M examples for SFT. Both stages share the same core optimization recipe for consistency, with minor stage-specific adjustments described in Appendix E.4.

## E.2. Data Composition

**Adapter pre-training.** We employ a warm-up strategy on the first 10% of pre-training data, mainly consisting of high-quality image-caption pairs and basic visual question-answering (VQA) tasks. The remaining 90% focuses on *General Knowledge Injection*, which draws from multiple public datasets, including InternVL-Chat-V1-2-SFT-Data, GRIT, LLaVAR, A-OKVQA, geo170K, LNQA, MAVIS, Screen2Words, and MMDU. The category composition of this subset is shown in

---

[1]Due to licensing and privacy constraints, the internal portion cannot be released. We provide the composition, task taxonomy, and full evaluation protocol to facilitate reproducibility of trends.

Figure 7(a).

**Supervised fine-tuning.** For the SFT stage, we curate a task-specific dataset with a balanced distribution of task types (Figure 7(b)), covering OCR, document understanding, captioning, math-centric multimodal reasoning, and pure-text instruction-following.

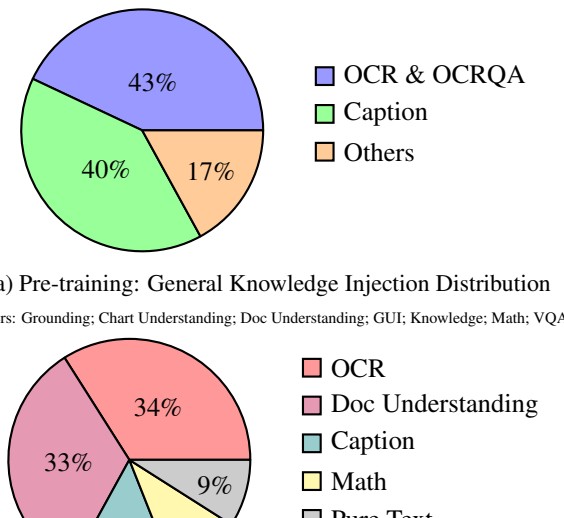

(a) Pre-training: General Knowledge Injection Distribution

Others: Grounding; Chart Understanding; Doc Understanding; GUI; Knowledge; Math; VQA

(b) SFT: Task Distribution

*Figure 7.* Data distribution of the pre-training and SFT stages.

### E.3. Optimization and Systems

We use the same optimizer and training infrastructure for both stages unless stated otherwise. The common settings are:

- **Optimizer:** AdamW

- **Weight decay:** 0.05

- **Warmup ratio:** 0.03

- **LR schedule:** Cosine decay

- **Precision and acceleration:** bf16 training, DeepSpeed ZeRO-3, FlashAttention-2

- **Global batch size:** 256

### E.4. Stage-specific Settings

**Adapter pre-training.** We freeze the parameters of the ViT and the LLM, and optimize only the adapter modules. We set the learning rate to $2 \times 10^{-5}$ and use a sequence length of 16k.

**Supervised fine-tuning (SFT).** For OCR- and VQA-centric tasks, we perform full-parameter fine-tuning following the same training recipe, with the following adjustments: we set the learning rate to $1 \times 10^{-5}$ and use a sequence length of 8k.

## F. Benchmarks and Evaluation Protocol

### F.1. Benchmark Suite

We evaluate our models on a suite of 22 benchmarks using a customized evaluation pipeline based on VLMEVALKIT. For systematic analysis, we group all benchmarks into four categories:

**STEM Puzzle.** MMMU$_{val}$, MathVista$_{mini}$, ScienceQA$_{TEST}$, ScienceQA$_{VAL}$.

**General.** A-Bench$_{VAL}$, CCBench, SEEDBench$_{IMG}$, SEEDBench2$_{Plus}$, MMVet, MMStar, BLINK, RealWorldQA.

**Alignment.** HallusionBench, POPE.

**OCR.** DocVQA$_{test}$, AI2D$_{test}$, ChartQA$_{test}$, OCRBench$_{div10}$, CharXiv$_{DQ}$, CharXiv$_{RQ}$, OCRVQA$_{testscore}$, TextVQA$_{val}$.

### F.2. Evaluation Pipeline

We adopt an automatic evaluation pipeline built upon VLMEVALKIT, with minor adaptations to match our model I/O format and to unify prompting templates across benchmarks. Unless a benchmark provides an official evaluation server, we follow the official released splits and compute the corresponding metrics locally.

**Prompting and formatting.** We use a unified prompt wrapper with a task-agnostic system instruction and a single-turn user query containing the image and the benchmark-specific question. For multiple-choice benchmarks, we format the candidate options verbatim and instruct the model to output the option letter only. For open-ended benchmarks, we instruct the model to output a concise final answer without additional explanations unless the benchmark explicitly requires rationales. All prompts and answer parsers used in our evaluation are included in the supplementary code release.

**Decoding.** We use greedy decoding (temperature $= 0$) by default to reduce evaluation variance. When a benchmark requires longer generations (e.g., certain reasoning-style QA), we increase the maximum generation length accordingly while keeping other decoding parameters unchanged. We apply no external tools (e.g., OCR engines, retrieval, or calculators) during evaluation unless the benchmark protocol explicitly assumes them.

### F.3. Per-benchmark Results

For completeness, we provide the full per-benchmark scores of all compared methods in Appendix G. These tables include (i) the raw score on each of the 22 benchmarks, (ii) the category-level macro averages reported in the main text, and (iii) the final overall average aggregated across all 22 benchmarks (rounded to one decimal place), and (iv) any benchmark-specific evaluation notes (e.g., answer normalization rules).

## G. Full Results

### G.1. Comparison

We provide detailed per-benchmark results to complement the category-level averages in Table 9. The table shows that gradient detachment consistently benefits OCR-centric benchmarks while preserving general reasoning performance.

### G.2. Eval on Different ViTs

*Table 9.* Detailed performance comparison across all benchmarks. We report results for **PE-baseline**, our proposed **Ours** (PE-best), and other fusion strategies including DenseConnector (DC), Multi-Layer Fusion (ML), and DeepStack. For **DC** and **ML**, we also show results with our detached gradient strategy applied (-detached). All scores are rounded to one decimal place.

| Benchmark | PE-baseline | Ours | DC | DC-detached | ML | ML-detached | DeepStack |
|---|---|---|---|---|---|---|---|
| **STEM Puzzle** | | | | | | | |
| STEM Puzzle Avg | 63.0 | 64.1 | 63.2 | **64.2** | 63.5 | 63.1 | 63.8 |
| MMMU$_{val}$ | 41.0 | 42.8 | 42.3 | **43.3** | 42.6 | 42.0 | 43.0 |
| MathVista$_{mini}$ | 43.3 | **45.2** | 44.3 | 45.1 | 44.9 | 44.4 | 44.8 |
| ScienceQA$_{TEST}$ | 83.9 | 83.9 | 82.9 | **84.2** | 83.0 | 82.8 | 83.6 |
| ScienceQA$_{VAL}$ | 83.8 | **84.7** | 83.5 | 83.9 | 83.4 | 83.2 | 84.0 |
| **General** | | | | | | | |
| General Avg | 53.2 | **54.6** | 54.0 | 54.4 | 54.1 | 54.0 | 54.5 |
| A-Bench$_{VAL}$ | 72.2 | 72.6 | 72.3 | 72.6 | 72.0 | 72.4 | **72.9** |
| CCBench | 38.6 | **39.4** | 39.0 | **39.4** | 38.2 | 38.3 | 38.9 |
| SEEDBench$_{IMG}$ | 71.3 | **72.5** | 72.1 | 72.0 | 72.3 | 72.0 | 72.0 |
| SEEDBench2$_{Plus}$ | 59.4 | **60.2** | 59.5 | 59.9 | 60.0 | 59.8 | 60.1 |
| MMVet | 40.8 | **42.2** | 41.0 | **42.2** | 41.8 | 41.2 | 42.0 |
| MMStar | 46.7 | 46.7 | 46.9 | 46.1 | 46.9 | **47.0** | 46.9 |
| BLINK | 43.7 | **47.3** | 46.2 | 46.8 | 47.1 | 47.0 | 47.2 |
| RealWorldQA | 52.7 | **56.1** | 55.1 | **56.1** | 54.3 | 54.6 | 56.0 |
| **Alignment** | | | | | | | |
| Alignment Avg | 72.6 | **73.6** | 72.5 | 72.8 | 72.6 | 73.2 | 73.2 |
| HallusionBench | 59.0 | **60.4** | 59.0 | 59.0 | 58.9 | 59.8 | 59.6 |
| POPE | 86.2 | **86.9** | 86.0 | 86.5 | 86.2 | 86.6 | 86.7 |
| **OCR** | | | | | | | |
| OCR Avg | 65.2 | **68.3** | 66.7 | 67.6 | 66.9 | 68.1 | 67.6 |
| DocVQA$_{test}$ | 71.4 | **73.2** | 73.0 | 72.9 | 72.6 | 73.1 | 72.9 |
| AI2D$_{test}$ | 71.9 | 73.3 | 73.6 | **73.8** | 73.5 | 73.2 | 73.6 |
| ChartQA$_{test}$ | 70.0 | **74.2** | 72.3 | 73.8 | 72.1 | 72.2 | 73.8 |
| OCRBench | 714 | 731 | 701 | 711 | 723 | **735** | 718 |
| CharXiv$_{DQ}$ | 69.7 | **73.0** | 70.3 | 71.2 | 72.0 | 72.3 | 70.9 |
| CharXiv$_{RQ}$ | 43.9 | 45.2 | 44.7 | 45.3 | 45.2 | **45.6** | 45.0 |
| OCRVQA$_{testscore}$ | 55.0 | 62.8 | 58.9 | 61.3 | 59.4 | **63.3** | 62.8 |
| TextVQA$_{val}$ | 68.7 | **71.3** | 70.3 | 71.1 | 68.0 | **71.3** | 69.8 |
| Overall Avg | 61.1 | **63.0** | 62.0 | 62.6 | 62.1 | 62.6 | 62.7 |

We report results across four ViT backbones to assess architectural generality in Table 10. Consistent gains across backbones indicate that Detached Skip-Links are robust to different visual encoders.

*Table 10.* Detailed ablation results across different ViT backbones: **PE**, **InternViT**, **AimV2**, and **SigLip2**. For each backbone, we compare the **Baseline** with our proposed Detached Skip-Links method (**Ours**). All scores are rounded to one decimal place.

| Benchmark | PE | | InternViT | | AimV2 | | SigLip | |
|---|---|---|---|---|---|---|---|---|
| | Baseline | Ours | Baseline | Ours | Baseline | Ours | Baseline | Ours |
| **STEM Puzzle** | | | | | | | | |
| STEM Puzzle Avg | 63.0 | 64.1 | 58.7 | 61.1 | 60.5 | 62.2 | 57.1 | 59.7 |
| MMMU$_{val}$ | 41.0 | 42.8 | 39.0 | 40.7 | 40.2 | 41.5 | 37.5 | 40.8 |
| MathVista$_{mini}$ | 43.3 | 45.2 | 38.6 | 40.3 | 40.5 | 42.1 | 37.8 | 39.2 |
| ScienceQA$_{TEST}$ | 83.9 | 83.9 | 78.0 | 82.1 | 80.5 | 82.5 | 76.3 | 79.4 |
| ScienceQA$_{VAL}$ | 83.8 | 84.7 | 79.0 | 81.3 | 80.8 | 82.9 | 76.8 | 79.3 |
| **General** | | | | | | | | |
| General Avg | 53.2 | 54.6 | 49.0 | 50.8 | 51.0 | 52.7 | 47.4 | 49.8 |
| A-Bench$_{VAL}$ | 72.2 | 72.6 | 69.6 | 71.3 | 70.8 | 71.9 | 62.5 | 65.9 |
| CCBench | 38.6 | 39.4 | 24.9 | 27.3 | 33.5 | 36.5 | 27.8 | 28.6 |
| SEEDBench$_{IMG}$ | 71.3 | 72.5 | 67.7 | 69.5 | 69.5 | 70.8 | 67.2 | 69.1 |
| SEEDBench2$_{Plus}$ | 59.4 | 60.2 | 61.5 | 61.7 | 58.5 | 60.1 | 57.7 | 59.6 |
| MMVet | 40.8 | 42.2 | 34.9 | 35.3 | 37.6 | 39.4 | 35.8 | 37.6 |
| MMStar | 46.7 | 46.7 | 42.9 | 46.2 | 44.5 | 46.5 | 39.9 | 43.9 |
| BLINK | 43.7 | 47.3 | 39.2 | 40.8 | 41.5 | 42.8 | 37.1 | 41.6 |
| RealWorldQA | 52.7 | 56.1 | 51.1 | 54.4 | 52.5 | 53.8 | 50.7 | 52.6 |
| **Alignment** | | | | | | | | |
| Alignment Avg | 72.6 | 73.6 | 64.0 | 71.4 | 69.5 | 71.5 | 69.6 | 69.4 |
| HallusionBench | 59.0 | 60.4 | 48.3 | 57.9 | 54.5 | 57.2 | 50.7 | 50.4 |
| POPE | 86.2 | 86.9 | 79.6 | 84.9 | 84.5 | 85.8 | 88.5 | 88.4 |
| **OCR** | | | | | | | | |
| OCR Avg | 65.2 | 68.3 | 60.3 | 62.2 | 63.0 | 64.8 | 58.1 | 60.6 |
| DocVQA$_{test}$ | 71.4 | 73.2 | 71.4 | 73.2 | 71.8 | 73.2 | 65.8 | 68.4 |
| AI2D$_{test}$ | 71.9 | 73.3 | 68.6 | 71.0 | 71.2 | 72.4 | 69.3 | 72.5 |
| ChartQA$_{test}$ | 70.0 | 74.2 | 73.3 | 74.9 | 69.5 | 72.0 | 73.5 | 75.2 |
| OCRBench | 714 | 731 | 656 | 664 | 685 | 702 | 608 | 641 |
| CharXiv$_{DQ}$ | 69.7 | 73.0 | 57.7 | 63.7 | 64.5 | 67.5 | 51.4 | 53.4 |
| CharXiv$_{RQ}$ | 43.9 | 45.2 | 43.0 | 43.3 | 43.5 | 44.1 | 36.2 | 37.6 |
| OCRVQA$_{testscore}$ | 55.0 | 62.8 | 38.3 | 39.5 | 48.5 | 51.5 | 41.9 | 44.6 |
| TextVQA$_{val}$ | 68.7 | 71.3 | 64.6 | 66.0 | 66.2 | 67.8 | 65.6 | 69.4 |
| Overall Avg | 61.1 | 63.0 | 56.2 | 58.7 | 58.8 | 60.6 | 55.0 | 57.3 |

