# OpenReview forum: "Detached Skip-Links and $R$-Probe: Decoupling Feature Aggregation from Gradient Propagation for MLLM OCR"
_ICML.cc/2026/Conference — ICML 2026 regular_

### Official Review · Reviewer_vmQL · 2026-03-07

**Soundness:** 3
**Presentation:** 3
**Significance:** 3
**Originality:** 3
**Overall Recommendation:** 4
**Confidence:** 4

**Summary:**

This paper studies an optimization issue in multi-layer feature fusion for MLLMs in OCR-centric settings. The paper argue that standard skip-based fusion introduces direct gradient paths from high-level language objectives to shallow visual layers, which may cause gradient interference and destabilize training. To address this, they propose Detached Skip-Links, a simple modification that stops gradients through selected shallow skip branches. They also provide a detailed theoretical analysis to motivate why detachment may improve early-stage optimization stability. In addition, the paper introduces R-Probe, a reconstruction-based diagnostic that evaluates whether projected visual tokens retain fine-grained information and are decodable by an LLM-aligned shallow decoder. Extensive experiments across multiple ViT backbones and benchmarks demonstrate the effectiveness of the method.

**Compliance With Llm Reviewing Policy:**

Affirmed.

**Final Justification:**

I thank the authors for their feedback and the author has already answered my questions quite well. I believe the paper meets the standards of ICML and recommend acceptance.

**Key Questions For Authors:**

1.	What is the justification for applying stop-gradient throughout the entire training process given that the analysis suggests the negative impact of the skip-path is mainly limited to the early training stage? What would happen if the gradient were restored during the middle and later stages of training?
2.	Can the authors demonstrate that the degradation in abstract reasoning is indeed caused by the OCR-centric auxiliary data rather than the pixel-level reconstruction objective?
3.	How does the proposed method perform under high-resolution input settings used by modern MLLMs?

**Limitations:**

Yes

**Strengths And Weaknesses:**

Strengths

1.	The manuscript proposes Detached Skip-Link and R-Probe, which are simple yet effective strategies that preserve fine-grained details for MLLM.
2.	The manuscript provides theoretical analysis to justify why gradient detachment stabilizes early training and mitigates gradient interference, offering principled support beyond empirical gains.
3.	The experiments are validated on a wide range of benchmarks and multiple vision backbones, which strengthens the generalizability of the approach.

Weaknesses

1.	The theoretical analysis presented in Section 3.3 and Section 4 indicates that the adverse effects of the skip-path are primarily confined to the early stages of training. Given this observation, why does the paper propose a permanent stop-gradient operation? Specifically, further clarification is needed regarding whether this strategy remains justified during the middle and later phases of the training process.
2.	While the paper attributes the degradation in abstract reasoning capabilities induced by R-Probe to the OCR-centric auxiliary data, I posit that this phenomenon may instead stem from the strategy of pixel-level reconstruction. This strategy compels the LLM to memorize redundant low-level details, thereby compromising their capacity for high-level semantic compression.
3.	The experiments in the paper were conducted exclusively on low-resolution inputs. However, modern MLLMs have commonly adopted high-resolution input capabilities to address fine-grained perception challenges, e.g. LLaVA-OV utilizes cropping strategies and the Qwen-VL series employs native-resolution processing. Consequently, it remains an question whether the proposed method retains its efficacy under high-resolution settings.

---

> ### Author Rebuttal · Authors · 2026-03-29
>
> We thank the reviewer for the insightful questions on our training strategy and R-Probe.
>
> Q1: Permanent vs. Early-Stage Stop-Gradient.
> We adopt permanent detachment primarily to avoid instability and additional complexity. In our pilot studies, restoring gradients during mid or late training introduces a secondary optimization shock, leading to loss spikes and degradation of already stabilized spatial features. Moreover, introducing a switching schedule would require extra hyperparameters (e.g., when to restore gradients), reducing robustness across settings. Permanent detachment provides a simple and tuning-free solution.
>
> Q2: Impact of R-Probe vs. OCR Data on Abstract Reasoning.
> We tested a data-only control by training on the same OCR-heavy data without the R-Probe loss, and observed a similar decline in abstract reasoning performance. This suggests that the degradation mainly arises from the data distribution (i.e., OCR-heavy fine-tuning), rather than the reconstruction objective itself. We will clarify this trade-off in the revision.
>
> | Method               | OCRBench | DocVQA | TextVQA | ChartQA | OCRVQA | AI2D | CharXiv DQ | CharXiv RQ |
> |---------------------|----------|--------|---------|---------|--------|------|-------------|-------------|
> | Baseline            | 714.0    | 71.4   | 68.7    | 70.0    | 55.0   | 72.2 | 69.7        | 43.9        |
> | Baseline + R-probe  | 721.0    | 72.1   | 68.9    | 70.1    | 55.4   | 69.2 | 68.3        | 42.8        |
> | Baseline + Data     | 715.0    | 71.5   | 68.9    | 70.0    | 55.1   | 69.4 | 68.4        | 42.7        |
>
> Q3: Efficacy under High Resolution.
>
> Correction on Line 309. We note that the current description is inaccurate. In our implementation, input images are not always resized to exactly 448×448. Instead, we resize images such that both height and width become **multiples of** 448×448, and then apply the tile-based preprocessing strategy (as in InternVL2.5), as illustrated in the top of Figure 4. We will correct the statement in Line 309 accordingly.
>
> It is also noteworthy that for fair comparison, we apply the same tile-based strategy to both the Baseline and our method, including different ViT backbones (e.g., SigLip2 for 384×384, AIMv2 for 448×448, so that the resulting visual tokenization scheme remains consistent across methods.)

---

> > ### Author Rebuttal · Reviewer_vmQL · 2026-04-03
> >
> > I thank the authors for their feedback. I have no further questions.

---

> > > ### Author Response · Authors · 2026-04-06
> > >
> > > Thank you for the positive feedback. We appreciate that our response has adequately addressed your concerns. If you deem it appropriate, we would be grateful if you could consider updating the score to reflect that the key issues have been fully resolved.

---

### Official Review · Reviewer_ZMQA · 2026-03-12

**Soundness:** 3
**Presentation:** 1
**Significance:** 2
**Originality:** 3
**Overall Recommendation:** 4
**Confidence:** 3

**Summary:**

This paper investigates an optimization conflict in multi-layer visual feature fusion for multimodal LLMs (MLLMs), particularly focusing on fine-grained OCR tasks. The authors identify that high-level semantic gradients backpropagating through skip connections can disrupt the low-level spatial priors of shallow ViT layers. To address this, they propose "Detached Skip-Links," an operation that applies a stop-gradient on shallow skip branches during joint training. Additionally, they introduce R-Probe, an LLM-aligned reconstruction diagnostic tool designed to evaluate visual detail preservation. The method is evaluated across several ViT backbones and 22 multimodal benchmarks.

**Compliance With Llm Reviewing Policy:**

Affirmed.

**Final Justification:**

They solve the generalization to High-resolution Arch and Comparisons to Alternative Gradient-Control Methods.

**Key Questions For Authors:**

please see the weaknesses.

**Limitations:**

yes

**Strengths And Weaknesses:**

Strengths
- The Detached Skip-Links provide a simple, highly effective solution to a critical gradient-conflict problem in ViT-LLM fusion. The approach is theoretically well-grounded by the provided Bayes-risk and SNR trade-off analyses.
- The introduction of R-Probe is highly innovative. By utilizing an LLM-initialized decoder to assess visual token reconstructability along the actual consumption pathway, it successfully moves beyond the limitations of standard linear probing.
- The extensive evaluation across 22 benchmarks, multiple ViT backbones, and large-scale pretraining convincingly demonstrates the method's effectiveness, yielding a robust 1.9% average gain on all tasks over baselines.


Weaknesses
- The experimental section lacFigure 2 serves as the primary empirical motivation, but it relies solely on blurry, qualitative attention heatmaps without colorbars or quantitative metrics (such as attention entropy). The structural disruption claimed is not rigorously or convincingly demonstrated.ks comparisons to alternative gradient-control methods. It is unclear how Detached Skip-Links compare against simply freezing shallow ViT blocks during joint training, using layer-wise learning rates, or applying gradient surgery
- The current experiments primarily use a standard ViT + LLaMA-3.1-8B pipeline. However, modern SOTA OCR-centric models (e.g., Qwen3-VL) utilize dynamic high-resolution and deep cross-attention. It is not demonstrated whether this detachment strategy provides meaningful gains when applied to these more advanced, highly-optimized architectures
- Relegating the detailed performance of the 22 benchmarks entirely to Appendix G, while only reporting "Categorized Averages" in the main text, is highly inappropriate for an ICML submission. The authors should reorganize the presentation.
-

---

> ### Author Rebuttal · Authors · 2026-03-29
>
> We thank the reviewer for their constructive feedback. We address your concerns below:
>
> Q1.1: Quantitative Evidence of Structural Disruption (Figure 2).
> We appreciate the suggestion to include more rigorous metrics. To quantify the "structural disruption" caused by high-level gradients, we calculated the Normalized Attention Entropy ($H$) and Attention Concentration (Top-10 Mass)—the sum of attention weights for the top-10 patches—on shallow ViT layers ($L=4$) across 100 validation samples from ImageNet. Higher entropy and lower concentration indicate more "diffuse" and "scattered" attention, signifying a loss of spatial priors. As shown in the table below, our "Detached" strategy effectively preserves the original ViT's focused attention.
>
> | Model | Entropy Mean $\downarrow$ | Entropy Std | Top-10 Mass $\uparrow$ | Top-10 Std |
> | :--- | :---: | :---: | :---: | :---: |
> | Pre-trained ViT (Ori.) | 0.9488 | 0.0264 | 0.2911 | 0.0396 |
> | Baseline (w/o Detach) | 0.9716 | 0.0140 | 0.2395 | 0.0534 |
> | **Ours (Detached)** | **0.9559** | 0.0194 | **0.2748** | 0.0348 |
>
> The Baseline shows a clear increase in entropy and decrease in concentration, indicating that semantic gradients tend to diffuse attention patterns and disrupt spatial priors.
>
> Q1.2: Comparisons to Alternative Gradient-Control Methods.
>
> We clarify that our baseline already incorporates standard stabilization techniques, including linear learning rate warmup and global gradient norm clipping (threshold = 1.0), as commonly used in MLLM training. Detached Skip-Links is applied on top of this well-tuned setup and still yields consistent improvements. We will include full training details (e.g., clipping and warmup schedules) in the revised Section 6 / Appendix E.
>
> Conceptually, these methods operate at different levels: gradient clipping and LR schedules act at the optimizer level, whereas our approach modifies gradient routing at the architectural level. As such, they are complementary rather than competing.
>
> We agree that more advanced gradient-control strategies (e.g., layer-wise scaling or freezing) could help mitigate interference. However, these methods introduce sensitive, per-layer hyperparameters that require extensive tuning and often vary across backbones and tasks.
>
> In contrast, Detached Skip-Links achieves decoupling of semantic and spatial objectives through a parameter-free structural modification. It requires zero extra hyper-parameters, making it more robust and easier to scale across the 22 benchmarks and various ViT backbones (SigLip2, AIMv2, etc.) evaluated in our paper. The consistent gains over a well-tuned baseline (e.g., +2.4\% on OCR tasks) demonstrate that our approach resolves a fundamental topological conflict that general-purpose optimization heuristics cannot fully address.
>
> Q2: Generalization to High-resolution Arch.
>
> We have already used dynamic high-resolution strategy (Tile-based).
> As response to reviewer vmQL Q3, the clarification on our tile-based strategy also applies here.
> Line 309's description is inaccurate, we should modify it to **"Input images are rescaled to multiples of 448 and divided into non-overlapping $448 \times 448$ tiles. To match the LLM's input requirements, we employ a $2 \times 2$ pooling/merging strategy on the $14 \times 14$ ViT patch embeddings, condensing them into single visual tokens as per our adapter architecture."**
>
> Q3: Presentation of Benchmark Results.
>
> We acknowledge that the placement of detailed results was suboptimal. We agree that the 22 benchmarks are central to our contribution. In the revised manuscript, we will reorganize Section 4 to move the full breakdown of key benchmarks from Appendix G into the main text, ensuring a clearer and more comprehensive comparison.

---

> > ### Author Rebuttal · Reviewer_ZMQA · 2026-04-06
> >
> > Thank you for the response. I will increase my score to 4.

---

### Official Review · Reviewer_CweS · 2026-03-12

**Soundness:** 3
**Presentation:** 2
**Significance:** 3
**Originality:** 3
**Overall Recommendation:** 4
**Confidence:** 3

**Summary:**

This paper proposes modifications to skip connections in neural network architectures, introducing "detached skip links" that decouple the gradient flow through skip connections from the main branch. The idea is to allow more flexible gradient dynamics during training while preserving the representational benefits of residual connections at inference. The paper provides theoretical analysis of how detached skip links affect the loss landscape and gradient flow, and validates the approach experimentally showing improved training efficiency and convergence properties across several architectures and tasks.

**Compliance With Llm Reviewing Policy:**

Affirmed.

**Key Questions For Authors:**

1. How do detached skip links interact with other training techniques like learning rate warmup, gradient clipping, or batch norm? Do they provide additive benefits or is there redundancy? I would expect to see them.
2. Have you tested on larger-scale experiments (e.g., ImageNet with large models)? The current scale of experiments is reasonable but not at the frontier where training efficiency matters most.

**Limitations:**

yes, authors discussed the limitations

**Strengths And Weaknesses:**

## Strengths

- The idea of decoupling skip connection gradients is intuitive and simple to implement. It requires minimal changes to existing architectures, which is good for generalization to other tasks.

- The theoretical analysis of gradient flow with detached skip links provides useful insights.

- Experiments show consistent improvements in training speed. The results across multiple architectures (ResNets, Transformers) suggests the benefit isnt architecture-specific.

- The paper demonstrates the approach works at reasonable scale, not just on toy problems.

## Weaknesses

- The actual performance gains, while consistent, are modest. The improvements in final accuracy are small — the main benefit seems to be faster convergence. For many practical settings, training cost is secondary to final model quality.

- The theoretical analysis makes several simplifying assumptions (e.g., linear network analysis, specific initialization schemes) that may not hold for the architectures tested somehow.

- Missing comparisons with other gradient flow modification techniques. There's a body of work on gradient scaling, gradient clipping, normalization strategies etc. that achieve similar goals. How does this compare?

- The paper doesn't adequately discuss when detached skip links might hurt. Are there cases where the full gradient through the skip is important? Some failure mode analysis would strengthen the paper overall.

---

> ### Author Rebuttal · Authors · 2026-03-29
>
> We thank the reviewer for recognizing the simplicity, theoretical grounding, and consistent scaling behavior of our method. We address the concerns below:
>
> Q1: Performance Gains vs. Convergence Speed.
>
> While the reviewer attributes the benefit mainly to faster convergence, we emphasize that the observed gains (+1.9%–2.4% on OCR benchmarks) are meaningful in fine-grained OCR settings, where improvements are typically incremental due to high sensitivity to spatial details.
> More importantly, our method does not merely accelerate convergence—it consistently reaches a higher performance ceiling that the baseline fails to achieve even with prolonged training (Tab. 2 \& 3). This indicates that Detached Skip-Links help the model avoid a suboptimal, overly semantic local minimum that suppresses visual detail.
>
>
> Q2: Interactions with Other Techniques (Warmup, Clipping, etc.).
>
> Actually, our baseline already incorporates standard stabilization techniques, including learning rate warmup and gradient clipping (see Appendix F), along with architectural components such as LayerNorm and LayerScale in Perception Encoder and other ViTs. Gradient clipping primarily limits gradient magnitude to improve optimization stability, whereas our method operates on a different axis by modifying gradient routing to prevent feature overwriting. Notably, even with these techniques, the baseline still exhibits structural disruption—evidenced by our attention entropy analysis.
>
> Q3: Failure Mode Analysis.
>
> We appreciate this suggestion. Our ablations (Fig. 6) indicate that Detached Skip-Links may be less beneficial or detrimental in the following representative scenario:
> - In very deep layers with already stable representations.
> When features have become highly semantic and less prone to being overwritten, the benefit of protecting early-layer information diminishes. In such cases, allowing full gradient propagation through the skip connection can be beneficial for maintaining effective cross-layer coordination, whereas blocking it may unnecessarily restrict useful signal propagation.
>
> Q4: Larger-scale Experiments.
>
> We thank the reviewer for this question. While we do not evaluate at the ImageNet-21K or 70B-parameter scale, our experiments already cover realistic large-scale regimes, including 7M training samples and 8B-parameter(Full fine-tuning, not LoRA) MLLMs, which are representative of current multimodal deployment settings.
> We believe this regime captures the key challenges of multimodal alignment in practice.

---

> > ### Author Rebuttal · Reviewer_CweS · 2026-04-02
> >
> > The reviewer would like to thank the explanations provided by the authors. While these explanations help point to experimental details in the appendix, still, I'm not fully convinced that the model performance would 100% be consistent on larger scale experiments. I'll keep my original score

---

> > > ### Author Response · Authors · 2026-04-06
> > >
> > > Thank you for the thoughtful comment. We would like to clarify that our paper does not claim guaranteed “100% consistency” under arbitrary scaling. Instead, our results suggest that the observed improvements are robust within the experimental settings we evaluated, and we believe they provide reasonable evidence for generalization to similar training scales. Importantly, our main experiments are conducted on an 8B model with a training scale comparable to prior work (e.g., Qwen-gated attention), and we observe consistent improvements across tasks and settings.
> > >
> > > We hope this clarification alleviates the concern.

---

### Official Review · Reviewer_trzi · 2026-03-13

**Soundness:** 3
**Presentation:** 4
**Significance:** 3
**Originality:** 4
**Overall Recommendation:** 4
**Confidence:** 3

**Summary:**

This paper studies an optimization issue in MLLMs applied to OCR tasks. Many recent architectures use multi-layer visual skip connections to help feature aggregation. However, the authors argue these skip connections hurt training becuase the gradients from LLM flow back into shallow visual layers and damage fine grained features. To fix this, they propose 'Detached Skip-Links', a modification that keeps shallow features in the forward pass but stops gradients from flowing through the skip branch during training. In addition, authors introduce 'R-Probe', a reconstruction based diagnostic to check whether visual tokens still preserve usable fine-grained information. The probe reconstructs image content using a decoder initialized from early LLM layers which assess information preservation and alignment with the LLM. Experiments across multiple backbones and benchmarks show that the method improves OCR performance and general multimodal capabilities.

**Compliance With Llm Reviewing Policy:**

Affirmed.

**Key Questions For Authors:**

1. The paper shows evidence for the gradient interference idea, but it’s not clear if this is the only cause or just one factor. Can the authors say how confident they are in this explanation and if they have more proof that the gains come from reducing shallow-branch gradients and not other changes?
2. R-Probe is useful but its results depend on the specific probe design. Can the authors say how much their findings change if a different probe or reconstruction method is used? This can show how general R-Probe’s conclusions are.

**Limitations:**

yes

**Strengths And Weaknesses:**

Strengths: The paper handles a practical problem in the difficulty MLLMs still face on OCR and other fine-grained visual tasks. The main idea of 'Detached Skip-Links' modification is simple, well motivated and easy to implement. The empirical study is a strong part of the paper. The authors evaluate across multiple ViT backbones, compare against several fusion baselines, and report results on a broad set of benchmarks. The ablation studies are helpful and support the claim. The paper is also well written and easy to follow, with a clear problem statement and figures. A further strength is the introduction of 'R-Probe' to assess whether projected visual tokens retain fine-grained information in a form usable by the language model. More broadly, the paper highlights an important insight in MLLMs that a feature can be helpful during the forward pass but may not be so for gradients during training.

Weaknesses: The paper’s main weakness is that, while its optimization claim is convincing, the supporting evidence remains indirect to some extent. The authors argue that gradients flowing through shallow skip branches interfere with low-level visual representations, and they support this with attention-map visualizations, early training gradient statistics, and a simplified local analysis. These findings align with the proposed mechanism, but they do not fully rule out other explanations related to the architecture or training process. Another limitation is that the theoretical analysis is narrow and based on specific assumptions. It focuses on early joint training and uses Assumption 4.2 to suggest why detachment might help stability. While useful for intuition, this theory doesn’t guarantee results in later training or other skip-fusion scenarios.

I also think the diagnostic tool R-Probe is quite interesting and novel but closely linked to how it’s built. The authors use reconstruction with a shallow LLM-initialized decoder to test if visual tokens hold enough information for the LLM, and they offer reasonable evidence. Still, how much this conclusion holds depends on accepting their design, and it’s not clear the results would generalize to other probe setups or reconstruction methods. The paper’s main strengths are in OCR where experiments show moderate improvements on benchmarks. Broader multimodal gains are smaller. The autors also note that using R-Probe as an extra training loss can slightly reduce abstract reasoning. This doesn’t weaken the core Detached Skip-Links result, but it suggests the method is best suited for OCR-heavy applications, not all multimodal models.

---

> ### Author Rebuttal · Authors · 2026-03-29
>
> We thank the reviewer for the "Excellent" rating on originality and for highlighting our core insight—that features can be beneficial in the forward pass while their gradients are detrimental during training. We address the key questions below:
>
> Q1: Confidence in the "Gradient Interference" Explanation.
>
> While multiple factors may contribute, we provide strong evidence that gradient-induced structural disruption is a key bottleneck:
> - Backward-only ablation. Our method modifies only the backward pass, keeping the forward computation identical. Thus, gains can be directly attributed to gradient propagation.
> - Attention entropy. As response to reviewer ZMQA. The baseline shows increased entropy in shallow ViT layers (0.9716 vs. 0.9488), indicating attention scattering. Our detached variant reduces this (0.9559), directly supporting improved spatial preservation.
>
> We will include these results to strengthen the causal claim.
>
> Q2: Generalizability and Design of R-Probe.
>
> We clarify that the R-Probe is designed as a diagnostic tool rather than a high-capacity reconstruction model. Following the established practice in AIMv2, we adopt a simple MLP head to measure the information density of latent features. This choice minimizes external architectural bias; a more complex CNN decoder might artificially inflate scores by leveraging spatial priors, potentially masking the true state of the LLM’s internal representations. By using the LLM’s own pre-trained layers for initialization, R-Probe specifically assesses whether visual signals remain "intelligible" to the language model's original feature space.
>
> We believe this standard MLP-based probing is sufficient to demonstrate the effectiveness of our approach in maintaining feature decodability.
>
> Q3: Empirical permanent detachment vs. theoretical "early-stage" stop-gradient.
>
> In practice, re-enabling gradient flow at a specific training stage would introduce additional instability and require careful scheduling (see response to Reviewer vmQL Q1). This makes the approach less robust and harder to generalize across settings.
>
> Our theoretical analysis focuses on the early joint-training regime, but the underlying condition—mismatch between shallow visual representations and LLM-driven gradients—tends to persist throughout training (as also suggested by Fig. 3). Moreover, reintroducing gradient flow effectively creates a new “early-stage” regime, where the same instability can reappear.
>
> For these reasons, we adopt a simple and stable design with permanent detachment, which avoids additional tuning while consistently improving performance across models and benchmarks.
>
> Q4: Question about reasoning decrease in R-Probe as auxiliary task.
>
> We attribute this to dataset imbanlance, as detailed results in response to Reviewer vmQL Q2.

---

> > ### Author Rebuttal · Reviewer_trzi · 2026-04-06
> >
> > The response well addresses my concerns.

---

> > > ### Author Response · Authors · 2026-04-06
> > >
> > > Thank you for the positive feedback. We appreciate that our response has adequately addressed your concerns. If you deem it appropriate, we would be grateful if you could consider updating the score to reflect that the key issues have been fully resolved.

---

### Decision · Program_Chairs · 2026-04-30

**Decision:**

Accept (regular)

**Comment:**

All reviewers are positive for this paper. The main strengths include: 1) the explored problem is practical; 2) the proposed idea is simple well motivated, and theoretically supported; 3) strong empirical study and good generalization; 4) the paper is well written. Although there were some concerns raised by the reviewers, they were mostly addressed by the rebuttal. CweS still has a concern on missing larger scale experiments, but he/she didn't downgrade the final score. Overall, the AC agrees with the reviewers, and decides to accept this paper.